# Juxtaposition of Bub1 and Cdc20 on phosphorylated Mad1 during catalytic mitotic checkpoint complex assembly

Elyse S. Fischer [1,4] ✉, Conny W. H. Yu[1,4], Johannes F. Hevler[2,3], Stephen H. McLaughlin [1], Sarah L. Maslen[1], Albert J. R. Heck [2,3], Stefan M. V. Freund[1] & David Barford [1] ✉

In response to improper kinetochore-microtubule attachments in mitosis, the spindle assembly checkpoint (SAC) assembles the mitotic checkpoint complex (MCC) to inhibit the anaphase-promoting complex/cyclosome, thereby delaying entry into anaphase. The MCC comprises Mad2:Cdc20:BubR1:Bub3. Its assembly is catalysed by unattached kinetochores on a Mad1:Mad2 platform. Mad1-bound closed-Mad2 (C-Mad2) recruits open-Mad2 (O-Mad2) through self-dimerization. This interaction, combined with Mps1 kinase-mediated phosphorylation of Bub1 and Mad1, accelerates MCC assembly, in a process that requires O-Mad2 to C-Mad2 conversion and concomitant binding of Cdc20. How Mad1 phosphorylation catalyses MCC assembly is poorly understood. Here, we characterized Mps1 phosphorylation of Mad1 and obtained structural insights into a phosphorylation-specific Mad1:Cdc20 interaction. This interaction, together with the Mps1-phosphorylation dependent association of Bub1 and Mad1, generates a tripartite assembly of Bub1 and Cdc20 onto the C-terminal domain of Mad1 (Mad1^CTD). We additionally identify flexibility of Mad1:Mad2 that suggests how the Cdc20:Mad1^CTD interaction brings the Mad2-interacting motif (MIM) of Cdc20 near O-Mad2. Thus, Mps1-dependent formation of the MCC-assembly scaffold functions to position and orient Cdc20 MIM near O-Mad2, thereby catalysing formation of C-Mad2:Cdc20.

During mitosis, kinetochores assembled on centromeric chromatin act as the interface for microtubule attachment to sister chromatids[1,2]. In response to improper attachments, anaphase onset is delayed by kinetochore-catalysed assembly of the mitotic checkpoint complex (MCC)[3–5]. The MCC binds and inhibits the anaphase-promoting complex/cyclosome (APC/C) whose activity triggers mitotic exit[6–9]. The MCC is composed of Mad2, Cdc20 and BubR1:Bub3[10,11]. The rate of Cdc20:Mad2 association poses a kinetic barrier to MCC formation[12,13].

This is overcome through a sequential assembly of checkpoint proteins onto the unattached outer kinetochore by means of an Mps1 kinase-dependent phosphorylation cascade that creates a catalytic scaffold for MCC formation[14,15] (Fig. 1a). Mps1 phosphorylates multiple MELT (Met-Glu-Leu-Thr) motifs of the outer kinetochore subunit Knl1 to control recruitment of Bub1:Bub3 and BubR1:Bub3[16–18]. Cdk1 and Mps1 then sequentially phosphorylate the conserved domain 1 (CD1) of Bub1 (Bub1^CD1), enabling it to bind the Mad1:Mad2 complex by

[1]MRC Laboratory of Molecular Biology, Cambridge Biomedical Campus, Francis Crick Avenue, Cambridge CB2 0QH, UK. [2]Biomolecular Mass Spectrometry and Proteomics, Bijvoet Center for Biomolecular Research and Utrecht Institute for Pharmaceutical Sciences, University of Utrecht, 3584 CH Utrecht, The Netherlands. [3]Netherlands Proteomics Center, University of Utrecht, 3584 CH Utrecht, The Netherlands. [4]These authors contributed equally: Elyse S. Fischer, Conny W. H. Yu. ✉e-mail: efischer@mrc-lmb.cam.ac.uk; dbarford@mrc-lmb.cam.ac.uk

**Fig. 1 | Assembly of the MCC at unattached kinetochores. a** The catalytic scaffold for MCC assembly onto the outer kinetochore. Phosphorylated Knl1 MELT motifs recruit the Bub3:Bub1 complex, which is sequentially phosphorylated by Cdk1 and Mps1 on the Bub1 CD1 domain. Phosphorylated Bub1 recruits the Mad1:C-Mad2 complex via its CD1 domain and O-Mad2 is recruited through self-dimerisation with C-Mad2. Mad1 is further phosphorylated by Mps1, which promotes interaction with Cdc20. **b** The catalytic scaffold shown in (**a**) functions to bring Cdc20 near O-Mad2. O-Mad2 undergoes conversion to C-Mad2, releasing it from Mad1:C-Mad2 and simultaneously binding Cdc20^MIM. C-Mad2:Cdc20 binds BubR1:Bub3 to generate the MCC. Once formed, soluble MCC binds and inhibits the APC/C, preventing anaphase progression. **c** Schematics of full-length Mad1, Bub1, Cdc20 and Mad2. The truncations used in this study are highlighted by the dashed boxes. MIM Mad2-

interacting motif. RLK Arg-Leu-Lys motif. RWD RING, WD40, DEAD domain. TPR tetratricopeptide repeat, GLEBS Gle2-binding-sequence, BDD Bub dimerisation domain, CD1 conserved domain 1, A1 ABBA motif, C1 C-box, K1/2 KEN-box motifs, IR IR-tail, HORMA Hop1p, Rev7p, Mad2 proteins. **d** A side view of the Mad1^485–718:C-Mad2:O-Mad2 complex. The Mad1 dimer encompassing residues 485-584 is depicted in orange, bound to two C-Mad2 molecules in light blue (PDB 1GO4)[23]. Two O-Mad2 molecules (dark blue) dimerised to C-Mad2 are fitted using the structure of the O-C Mad2 dimer (PDB 2V64)[26]. The coiled-coil is then interrupted by a 16-residue segment after which the coiled-coil is extended to the C-terminal domain of Mad1 (residue 597–718), which is depicted in orange (PDB 4DZO)[24]. The RLK motif with Mad1^CTD is highlighted in yellow.

interacting with the RLK (arginine-leucine-lysine) motif within the C-terminus of Mad1 (Mad1^CTD) (Fig. 1a, c, d)[14,19–22]. Mad1 self-dimerises through a series of coiled-coil α-helices. The coiled-coil formed from residues 485-584 is interrupted by the Mad2-interacting motif (MIM) that entraps one molecule of Mad2 per subunit to generate the

Mad1:C-Mad2 tetramer (Fig. 1d)[23]. This segment is followed by a flexible linker (residues 585–597), after which the coiled-coil resumes, ending in a globular head domain presenting an RWD-fold (residues 597–718) (Fig. 1d)[24]. During MCC assembly, O-Mad2 is targeted to the outer kinetochore through self-dimerisation to the Mad1-bound C-

Mad2 (Fig. 1b, d)[23,25,26]. A substantial remodelling of dimerised O-Mad2 into C-Mad2, comprising the C-terminal 'safety-belt', is then required for Mad2 to entrap the MIM of Cdc20[27–30] (Fig. 1b, c). The C-Mad2:Cdc20 complex has a high affinity for BubR1:Bub3, allowing spontaneous MCC assembly (Fig. 1b)[11,12,31].

In vitro, conversion of Mad2 takes several hours, whereas, in cells, the SAC response is established within minutes of kinetochore-microtubule detachment[12,13,29,32]. However, in the presence of components of the catalytic scaffold (Fig. 1a), nearly spontaneous conversion of Mad2 occurs in vitro[12,33]. Understanding how the identified catalysts (Mad1:C-Mad2, Bub1 and Mps1) promote Mad2 conversion and concomitant Cdc20 association is of key interest and remains largely unanswered. A critical requirement of a functional checkpoint in vivo, and catalytic MCC formation in vitro, is Mps1 phosphorylation of the Mad1 C-terminus[14,33–35] (Fig. 1c, d). Phosphorylated Mad1 interacts with a conserved basic motif at the N-terminus of Cdc20 (Box1) (Fig. 1c)[14,33]. Disrupting either phosphorylation of the Mad1 C-terminus, or abolishing the Cdc20:Mad1 interaction, does not prevent Cdc20 kinetochore localisation, but causes a defective checkpoint, and impairs catalytic Cdc20:C-Mad2 formation in vitro[14,33,35]. These findings led to the suggestion that the Cdc20:Mad1 interaction allows Cdc20 to be incorporated into the checkpoint by promoting both the accessibility of the MIM, and its presentation to Mad2 for safety-belt entrapment[33]. This mechanism is likely dependent on both relieving Cdc20 auto-inhibition and optimally positioning its MIM close to Mad2 as it undergoes conversion[33,35]. However, the molecular mechanisms for how Mad1 phosphorylation mediates interaction with Cdc20, and how this interaction promotes MIM accessibility and catalytic MCC formation are unknown.

In this study, we applied complementary approaches to investigate how Mad1 phosphorylation regulates its interaction with Cdc20, and how this interaction likely promotes Cdc20:Mad2 binding. We determined that the Cdc20:Mad1 interaction is modulated by the phosphorylation of Mad1 Thr716 (pThr716). Using NMR spectroscopy, we demonstrated that an N-terminal segment of Cdc20 binds across Mad1$^{CTD}$ through two regions with residual α-helicity (α1 and Box1). Biophysical characterisation and fluorescence anisotropy measurements identified that a single copy of both Bub1$^{CD1}$ and Cdc20$^N$ concurrently bind to a phosphorylated Mad1$^{CTD}$, and AlphaFold2 was used to predict an experimentally supported model of this tripartite assembly. Using a combination of cryo-EM and cross-linking mass spectrometry, we also identified a folded state of the Mad1:Mad2 complex. Together with the Cdc20:Mad1$^{CTD}$ interaction, this suggests a model for how the MIM of Cdc20 is presented to Mad2. Lastly, we used NMR to show that Cdc20$^{MIM}$ alone is sufficient to induce Mad2 conversion, highlighting the importance of the Mad1-Bub1 scaffold in orchestrating the arrangement of Cdc20 and Mad2 during catalytic MCC assembly.

## Results

### Thr716 is the predominant Mps1 phosphorylation site of Mad1$^{CTD}$

Several Mps1-phosphorylation sites within Mad1 have been proposed to enhance MCC formation[14,33–35]. However, both the specific function of each of these sites, and the molecular mechanism of how Mad1 phosphorylation promotes interaction with Cdc20 are unknown. We phosphorylated Mad1$^{CTD}$ with full-length Mps1 in vitro to identify phosphorylation sites on Mad1$^{CTD}$. The majority of phosphorylated peptides identified by mass spectrometry contain a phosphorylated Thr716 (pThr716) site, with a few being phosphorylated at Thr644 (Supplementary Table 1). Intact protein-centric mass spectrometry showed that each monomer of phosphorylated Mad1$^{CTD}$ and phosphorylated Mad1$^{485–718}$ contained only a single phosphorylated residue (Supplementary Fig. 1a, b). We further analysed phosphorylated Mad1$^{CTD}$ using NMR to monitor the previously assigned backbone resonances of unphosphorylated Mad1$^{CTD}$ in $^1$H, $^{15}$N 2D HSQC spectra[21]

(Supplementary Fig. 1c). Marked chemical shift perturbations (CSP) were only observed for the backbone resonances corresponding to Thr716 and its neighbouring residues (713–718), consistent with preferred Thr716 phosphorylation (Supplementary Fig. 1d). Our data indicate that in vitro Thr716 is the only highly occupied Mps1 phosphorylation site of Mad1$^{CTD}$. This agrees with previous reports that phosphorylation of Thr716 is critical for both maintaining a viable checkpoint in vivo[14,34], and the ability of the MCC to assemble and inhibit the APC/C in vitro[14].

### Mad1$^{CTD}$ Thr716 phosphorylation promotes the binding of a single copy of Cdc20

The N-terminus of Cdc20, specifically its Box1 motif, was proposed to interact with Mad1$^{CTD}$ in a phosphorylation-dependent manner[14,33]. We sought to investigate this interaction using Mad1$^{CTD}$ and an N-terminal truncation of Cdc20 (residues 1–73; Cdc20$^N$) (Fig. 1c). The interaction of Cdc20$^N$ and Mad1$^{CTD}$ was analysed by SEC-MALS. Although Cdc20$^N$ did not bind to unphosphorylated Mad1$^{CTD}$ (Fig. 2a), it formed a stable complex with Mad1$^{CTD}$ phosphorylated at Thr716 (pMad1$^{CTD}$) (Fig. 2b). This indicated a strong interaction between pMad1$^{CTD}$ and Cdc20$^N$, dependent on phosphorylation of Mad1 at Thr716. The elution peak corresponding to pMad1$^{CTD}$:Cdc20$^N$ was monodispersed with an observed mass of 38 kDa (Fig. 2b), close to the expected mass of a pMad1$^{CTD}$ dimer with one Cdc20$^N$ bound (36 kDa). AUC-SE was used to further investigate the stoichiometry of the Cdc20$^N$:pMad1$^{CTD}$ complex. These data also indicated that only a single copy of Cdc20$^N$ bound to the pMad1$^{CTD}$ homodimer (Supplementary Fig. 2), even with a four-fold molar excess of Cdc20$^N$ at concentrations well above the assumed $K_D$, judged by the strong interaction of Mad1$^{CTD}$ and Cdc20$^N$ in SEC, and later confirmed by fluorescent anisotropy experiments discussed below.

### Characterisation of the Cdc20$^N$:Mad1$^{CTD}$ interaction by NMR

NMR spectroscopy was used to obtain structural insights into the Cdc20$^N$:Mad1$^{CTD}$ interaction. Unlabelled Cdc20$^N$ was titrated into either $^{15}$N-labelled Mad1$^{CTD}$ or pMad1$^{CTD}$ (Fig. 2c, d and Supplementary Fig. 3a, b). Significant signal attenuation was observed in $^1$H, $^{15}$N 2D HSQC spectra for resonances corresponding to residues 616–660 of both Mad1$^{CTD}$ and pMad1$^{CTD}$ (Fig. 2e). Mapping these line broadened residues onto the Mad1$^{CTD}$ crystal structure (PDB 4DZO)[24] indicated that regions of Mad1$^{CTD}$ comprising the RLK motif, the coiled-coil, and first β-strand of the head domain, were perturbed upon Cdc20$^N$ binding (Fig. 2f). Line broadening was more pronounced and extensive for pMad1$^{CTD}$, with signal attenuation observed in the entire β-sheet, the C-terminal α-helix, and the pThr716 site (Fig. 2g). Our results are consistent with a previous study which showed that mutation of the conserved QYRL motif (Q648A and R650A) within the Mad1$^{CTD}$ β-sheet, and an RLK/AAA (residues 617-619) mutation, impaired binding of Cdc20$^N$ to Mad1 (Fig. 2e–g)[33]. Overall, the extent of line broadening observed upon Cdc20$^N$ binding suggests that Cdc20$^N$ forms an extensive binding interface with Mad1$^{CTD}$, and/or it induces a global conformational change within Mad1$^{CTD}$.

The N-terminus of Cdc20 contains two conserved basic motifs, Box1 (27-RWQRKAKE-34), which is predicted to be α-helical, and Box2 (58-RTPGKSSSKVQT-69) which is predicted to be unstructured (Supplementary Fig. 4). Box1 has been implicated in binding pMad1$^{CTD}$ and found to be functionally important for checkpoint formation[14,33]. N-terminal to Box1 and Box2 is another predicted α-helix (Cdc20 α1) with an unknown function (Supplementary Fig. 4). To obtain structural information on Cdc20$^N$, and to reciprocally define Mad1$^{CTD}$-binding sites on Cdc20$^N$, we characterised isotopically labelled Cdc20$^N$ using NMR. The narrow dispersion of $^1$H chemical shifts indicated that Cdc20$^N$ was largely disordered (Supplementary Fig. 5a, b). Several backbone amide signals were missing in the $^1$H, $^{15}$N 2D HSQC spectrum comprising prolines and residues that underwent rapid exchange with

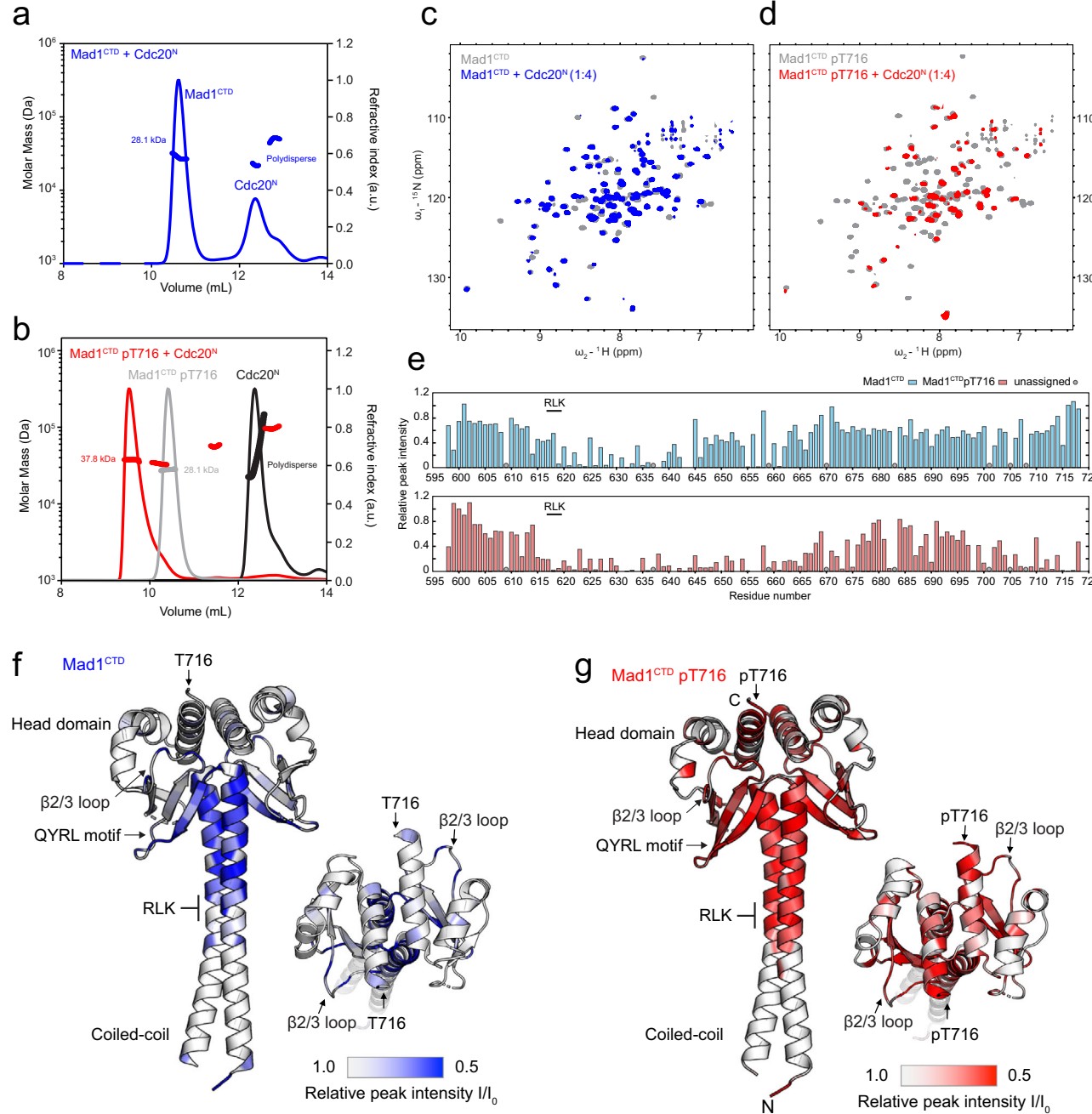

**Fig. 2 | A direct Cdc20-Mad1 interaction mediated by phosphorylated Mad1^CTD pThr716. a** SEC-MALS analysis of unphosphorylated Mad1^CTD + Cdc20^N (blue). The first elution peak consists of free Mad1^CTD (expected mass 28 kDa), while Cdc20^N (expected mass 7.8 kDa) alone elutes in the second polydispersed peak (later confirmed to be homogenous monomer by AUC-SE in Supplementary Fig. 2), suggesting there is no binding between unphosphorylated Mad1^CTD and Cdc20^N. **b** SEC-MALS analysis of free Mad1^CTD pT716 (grey), free Cdc20^N (black) and Mad1^CTD pT716 + Cdc20^N (red). Mad1^CTD pT716 alone (grey) eluted as one monodispersed species with an apparent mass of 28.1 kDa. Cdc20^N alone (black) elutes as a poly-dispersed peak with an undefined mass. Mad1^CTD pT716 + Cdc20^N (red) gives a distinct monodispersed species of 37.8 kDa, indicative of a stable complex with one Mad1^CTD pT716 dimer and a single molecule of Cdc20^N (expected 36 kDa). **c** ^1H,^15N 2D HSQC showing ^15N-labelled unphosphorylated Mad1^CTD alone (grey) or in the presence of four molar excess Cdc20^N (blue). **d** ^1H,^15N 2D HSQC showing ^15N-labelled

Mad1^CTD pT716 alone (grey) or in the presence of four molar excess Cdc20^N (red). **e** Relative peak intensities from the ^1H,^15N 2D HSQC of Mad1^CTD (blue) or Mad1^CTD pT716 (red) in the presence of Cdc20^N at 1:1 molar ratio were mapped onto the Mad1^CTD sequence. Peak intensities were normalised to the C-terminal residue Ala718 for the spectra of unphosphorylated Mad1^CTD, where the C-terminus was clearly not involved in binding. For phosphorylated Mad1^CTD the peak intensities were adjusted to match the spectra collected for unphosphorylated Mad1^CTD for comparison, as the experimental conditions were nearly identical. Unassigned peaks are denoted as grey circles. **f** Line broadening induced by Cdc20^N binding to unphosphorylated Mad1^CTD was mapped onto the structure of Mad1^CTD (PDB 4DZO)[24]. Blue in the colour scale indicates regions with significant line broadening. The RLK motif (617-619) and QYRL motif (648-651) are labelled. **g** Line broadening induced by Cdc20^N binding to Mad1^CTD pT716 was mapped onto the structure of Mad1^CTD. Red in the colour scale indicates regions with significant line broadening.

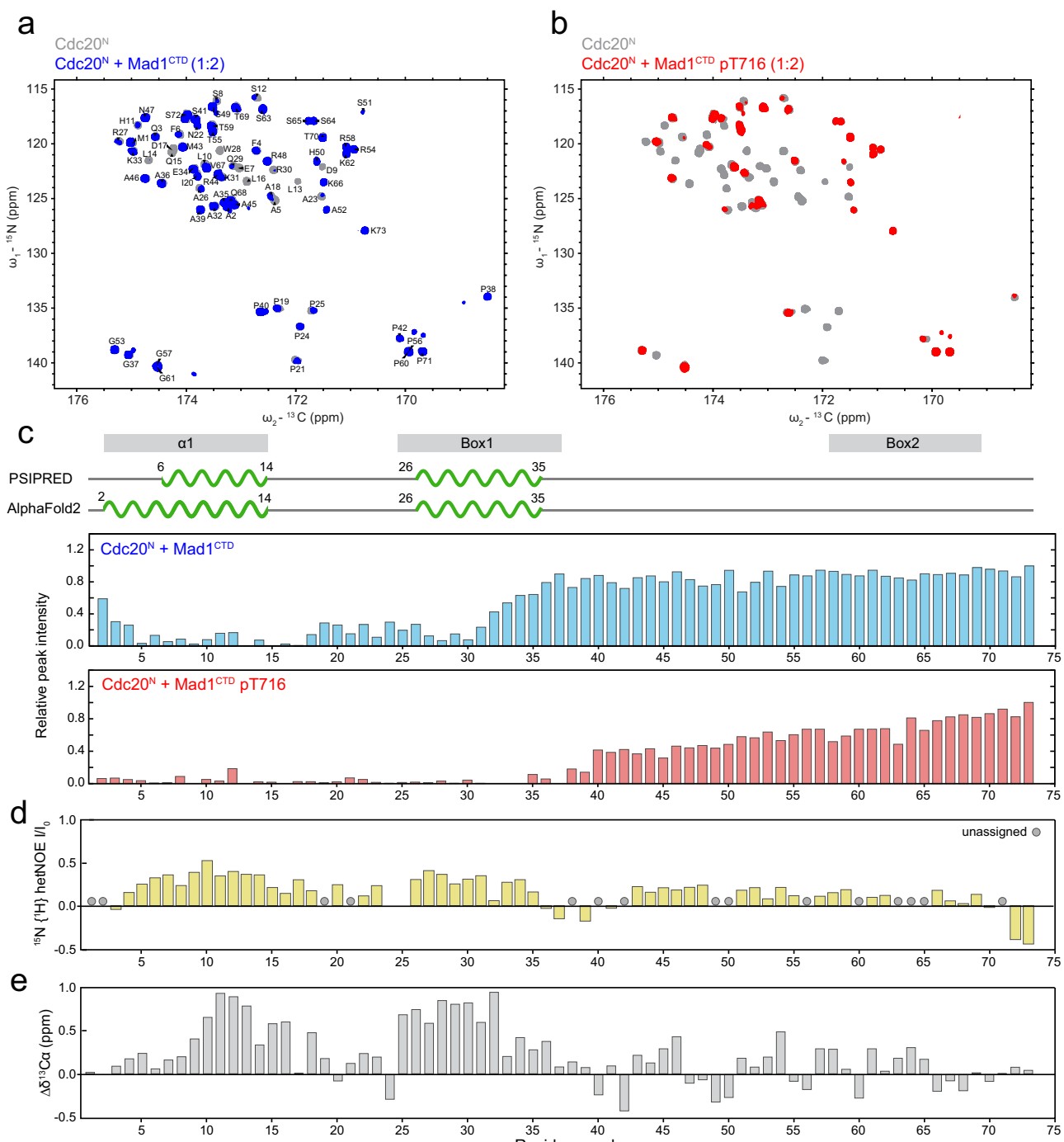

**Fig. 3 | Characterisation of Cdc20^N and its interaction with Mad1^CTD.**
**a** ^13C-detected 2D CON spectra of Cdc20^N alone (grey) and in the presence of Mad1^CTD (blue) at a four molar excess, shown with the assignment of Cdc20^N backbone resonances. **b** ^13C-detected 2D CON spectra of Cdc20^N alone (grey) and in the presence of Mad1^CTDpT716 (red) at a twofold molar excess. **c** Relative peak intensities from the CON spectra of Cdc20^N in the presence of Mad1^CTD (blue) and Mad1^CTD pT716 (red) were mapped onto the Cdc20^N sequence. Peak intensities were normalised to the C-terminal residue Gly73. Regions showing significant line broadening coincide with regions that were predicted to be α-helical by both PSIPRED and AlphaFold2, as shown in the schematics above. **d** ^15N{^1H}-hetero-nuclear NOE measuring the backbone dynamics in Cdc20^N and reported as mean values in technical duplicates. A higher (more positive) value indicates higher rigidity of the backbone amide bond, while a lower value indicates higher flexibility. **e** Secondary chemical shifts (Δδ^13Cα) reveal residual secondary structure in Cdc20^N. Positive values suggest α-helical propensity and negative values suggest a propensity for β-strand conformation.

the solvent (Supplementary Fig. 5a, b). To obtain a complete assignment of the backbone resonances of Cdc20^N we employed experiments that directly detect the ^13C nuclei (^13C-detected) to define residues based on their ^15N and ^13C carbonyl frequencies (Fig. 3a). For the same reason, we used ^13C-detected 2D CON spectra that report the correlation between backbone carbonyl ^13C and amide ^15N, to provide a complete picture of the effect of Mad1^CTD binding on Cdc20^N (Fig. 3a, b). The relative peak intensity changes were mapped onto the Cdc20^N sequence (Fig. 3c). When either phosphorylated or unphosphorylated Mad1^CTD were titrated, substantial signal attenuation in every residue of the N-terminal half of Cdc20^N was observed, whereas the C-terminal half including Box2 was largely unaffected. This indicated that both

Cdc20 α1 and Box1, but not Box2, are involved in Mad1[CTD] binding. Notably, four prolines located in the loop connecting Cdc20 α1 and Box1, are also involved in the interaction, suggesting these residues might provide some rigidity to orientate these α-helical segments. More significant signal attenuation occurred with pMad1[CTD] (Fig. 3 and Supplementary Fig. 5), confirming the importance of Mad1 phosphorylation in promoting this interaction.

We further delineated the binding sites of α1 and Box1, modelled as short peptides, on Mad1[CTD] by monitoring the chemical shift perturbations (CSPs) upon peptide titrations (Supplementary Figs. 3, 6). As expected, these CSPs were mapped to more localised regions within Mad1[CTD]. Cdc20 Box1 binds pMad1[CTD] but not Mad1[CTD] and CSPs were observed around pThr716, the β2 strand, and β2/β3 loop adjacent to pThr716 (residues 652-661, Fig. 2g and Supplementary Fig. 6b, c). Meanwhile, very small CSPs were observed when Cdc20 α1 was titrated to pMad1[CTD], even at high concentrations of peptide, suggesting a weak interaction between Cdc20 α1 and pMad1[CTD]. These CSPs were mapped to the β-sheet and proximal coiled-coil of pMad1[CTD] (Supplementary Fig. 6a). The low affinity of these interactions suggests that cooperative binding involving α1 and Box1 is required to create a stable interaction between Cdc20[N] and Mad1[CTD]. The interaction of Box1 close to pThr716 and Cdc20 α1 near the Mad1 RLK motif would suggest that Cdc20 binds Mad1[CTD] in a parallel orientation, and is in agreement with cross-linking mass spectrometry data that Cdc20[Box1] interacts with the head domain of Mad1[CTD] [ref. 33].

Although AlphaFold2 and PSIPRED predictions modelled Cdc20 as having two α-helical segments at its N-terminus (Supplementary Fig. 4), Cdc20[N] lacks a stable secondary structure in solution as indicated by its [1]H chemical shift dispersion (Supplementary Fig. 5a, b). The absence of a stable secondary structure is supported by [15]N{[1]H}-heteronuclear NOE experiments that sample [15]N backbone dynamics on a fast picosecond timescale. These showed only a slight decrease in flexibility for the backbone amides in the predicted α-helical regions compared with predicted disordered regions (Fig. 3d). To identify possible residual α-helicity in these regions, we examined the secondary chemical shifts of Cdc20[N] by comparing the differences in Cα chemical shifts between native and denatured states[36]. Secondary chemical shifts are a reliable measurement of secondary structure propensity, especially in disordered proteins[37]. The secondary chemical shifts of Cdc20[N] indicated a clear α-helical propensity for both α1 and Box1 (Fig. 3e). Substantial line broadening occurred when Cdc20[N] interacted with pMad1[CTD], preventing us from acquiring similar backbone dynamics and secondary chemical shift data on the pMad1[CTD]:Cdc20[N] complex. However, the residual α-helical propensity would be consistent with the idea that Cdc20 α1 and Box1 adopt an α-helical conformation upon binding to pMad1[CTD]. Altogether, our data suggest an extensive interaction between Mad1[CTD] and the N-terminal half of Cdc20[N].

## A tripartite assembly of Cdc20[N] and Bub1[CD1] on phosphorylated Mad1[CTD]

We previously showed that only a single Bub1[CD1] binds to the Mad1[CTD] homodimer, most likely a result of the inherent asymmetry within Mad1[CTD] in which the coiled-coil is bent with respect to the head domain, observed in both the Bub1[CD1] bound and apo-Mad1[CTD] X-ray structures in multiple different crystal lattices[21,24]. Interestingly, the Mad1[CTD] homodimer also binds only one Cdc20[N] (Fig. 2b and Supplementary Fig. 2). We reasoned that Mad1[CTD] asymmetry might also play a role in defining the Cdc20[N]:Mad1[CTD] stoichiometry. As shown by NMR, residues of Mad1[CTD] that bind Cdc20[N] almost entirely overlap in sequence with those that bind Bub1[CD1] (Fig. 2c–g)[21].

To investigate whether Cdc20[N] and Bub1[CD1] either compete for binding to pMad1[CTD], or bind concurrently, we measured the change in fluorescence anisotropy of a Bub1 peptide bound to Mad1[CTD] as a function of Cdc20[N] concentration. For this, we conjugated Aurora

Fluor 488 (AF488) to the flexible N-terminus of a phosphorylated Bub1[CD1] peptide. AF488-Bub1[CD1] and Mad1[CTD] formed a complex with a binding affinity of 0.7 μM for Mad1[CTD] and 0.9 μM for pMad1[CTD] (Fig. 4a). This agrees with the 2.7 μM affinity previously determined by ITC[21], indicating that the fluorescent tag does not interfere with Mad1[CTD] binding. The equivalent affinities of AF488-Bub1[CD1] for phosphorylated and unphosphorylated Mad1[CTD] are in agreement with our previous crystal structure, showing that pThr716 would not contact Bub1[CD1] [ref. 21]. Next, we titrated Cdc20[N] into the preformed AF488-Bub1[CD1]:pMad1[CTD] complex. In the case of competition, the introduction of unlabelled Cdc20[N] would displace AF488-Bub1[CD1], resulting in a decrease in fluorescence anisotropy as its tumbling motion increases in the unbound state. Instead, titration of Cdc20[N] into the AF488-Bub1[CD1]:Mad1[CTD] or AF488-Bub1[CD1]:pMad1[CTD] complex increased the fluorescence anisotropy (Fig. 4b). This indicated that Cdc20[N] and Bub1[CD1] bind to Mad1[CTD] at two distinct sites simultaneously. Cdc20[N] binds to the preformed Mad1[CTD]:AF488-Bub1[CD1] complex with a $K_D$ of 4.8 μM for pMad1[CTD] and 28 μM for Mad1[CTD] (Fig. 4b). We next determined the affinity of AF488-Bub1[CD1] for a preformed pMad1[CTD]:Cdc20[N] complex. This showed a similar affinity (1.7 μM) to the binding of AF488-Bub1[CD1] to pMad1[CTD] alone (0.9 μM), indicating that Cdc20 and Bub1 binding to pMad1[CTD] is unlikely to be cooperative (Fig. 4c). A low micromolar affinity of the Cdc20[N] interaction agrees with the timescales observed in NMR where line broadening of Mad1[CTD] peaks occurred upon Cdc20[N] binding (Fig. 2c, d). These affinities are also consistent with the stability of the complex observed in SEC when Mad1[CTD] is phosphorylated but not when Mad1[CTD] is unphosphorylated (Fig. 2a, b). Thus, our data indicate that Bub1[CD1] and Cdc20[N] can bind simultaneously to pMad1[CTD] to form a stable tripartite complex.

The pMad1[CTD]-Cdc20[N] complex failed to crystallise using a variety of Cdc20[N] peptides and supplementing Bub1[CD1]. We, therefore, used AlphaFold2[38] to predict the structure of the Cdc20[N]:Bub1[CD1]:Mad1[CTD] tripartite assembly. On imposing a 1:1:2 stoichiometry of Cdc20[N]:Bub1[448–550]:Mad1[CTD], AlphaFold2 predicted a tripartite complex with high confidence (>90 predicted local distance difference test (pLDDT) for Bub1[CD1] and 70-90 pLDDT for Cdc20[N]) (Fig. 4d and Supplementary Fig. 7a–c). The predicted aligned error (PAE) plot also indicates high confidence for the orientation of Bub1[CD1] and Cdc20[1–35] with respect to Mad1[CTD] (Supplementary Fig. 7d). A comparable tripartite complex was also predicted when full-length Bub1 and Cdc20 were used (data not shown). The predicted model of the Bub1[CD1]:Mad1[CTD] interaction was nearly identical to our Bub1[CD1]:Mad1[CTD] crystal structure (PDB: 7B1F)[21] and agreed with the binding sites mapped by NMR (Fig. 4f, g)[21], providing confidence in the AlphaFold2 model. The crystal structure of Mad1-Bub1[21] was used to build pSer459 and pThr461 of Bub1, and to model pThr716 of Mad1. For the Cdc20[N]:Mad1[CTD] interaction, AlphaFold2 predicts that the first 35 residues of Cdc20[N] are involved in the Cdc20:Mad1[CTD] interaction and α1 of Cdc20 runs parallel to the Mad1[CTD] coiled-coil, consistent with our NMR data (Figs. 3, 4h and Supplementary Fig. 6).

In this predicted model, the Cdc20[N] α1 helix lies diagonally across the Mad1[CTD] coiled-coil with its N-terminus close to the Mad1 RLK motif of one monomer, and the C-terminus contacting the head domain β-sheet of the adjacent monomer, in a manner that strikingly mimics the Bub1[CD1]:Mad1[CTD] interaction (Fig. 4d). Similar to Bub1[CD1], the interaction of Cdc20 α1 with Mad1[CTD] would largely be driven by hydrophobic interactions (Fig. 4e). This model predicts that Gln3 and Glu7, and Asp9 of Cdc20 α1 interact with Arg617 and Lys619, respectively of the Mad1[CTD] RLK motif (Fig. 4d). This is consistent with a previous report that a Mad1[CTD] RLK/AAA mutant abolished Cdc20 binding[33], and our NMR data where the RLK residues of Mad1[CTD] were perturbed upon Cdc20[N] binding (Fig. 2e). An interaction of Cdc20 Asp9 with Mad1 Lys619 might also explain how a Mad1[CTD] K619A mutant disrupted the checkpoint, even though the Mad1[CTD] K619A mutant did not disrupt Bub1[CD1] binding significantly[21,24]. To further test the interaction of Cdc20 α1 with the RLK motif of Mad1, we assessed the binding of a

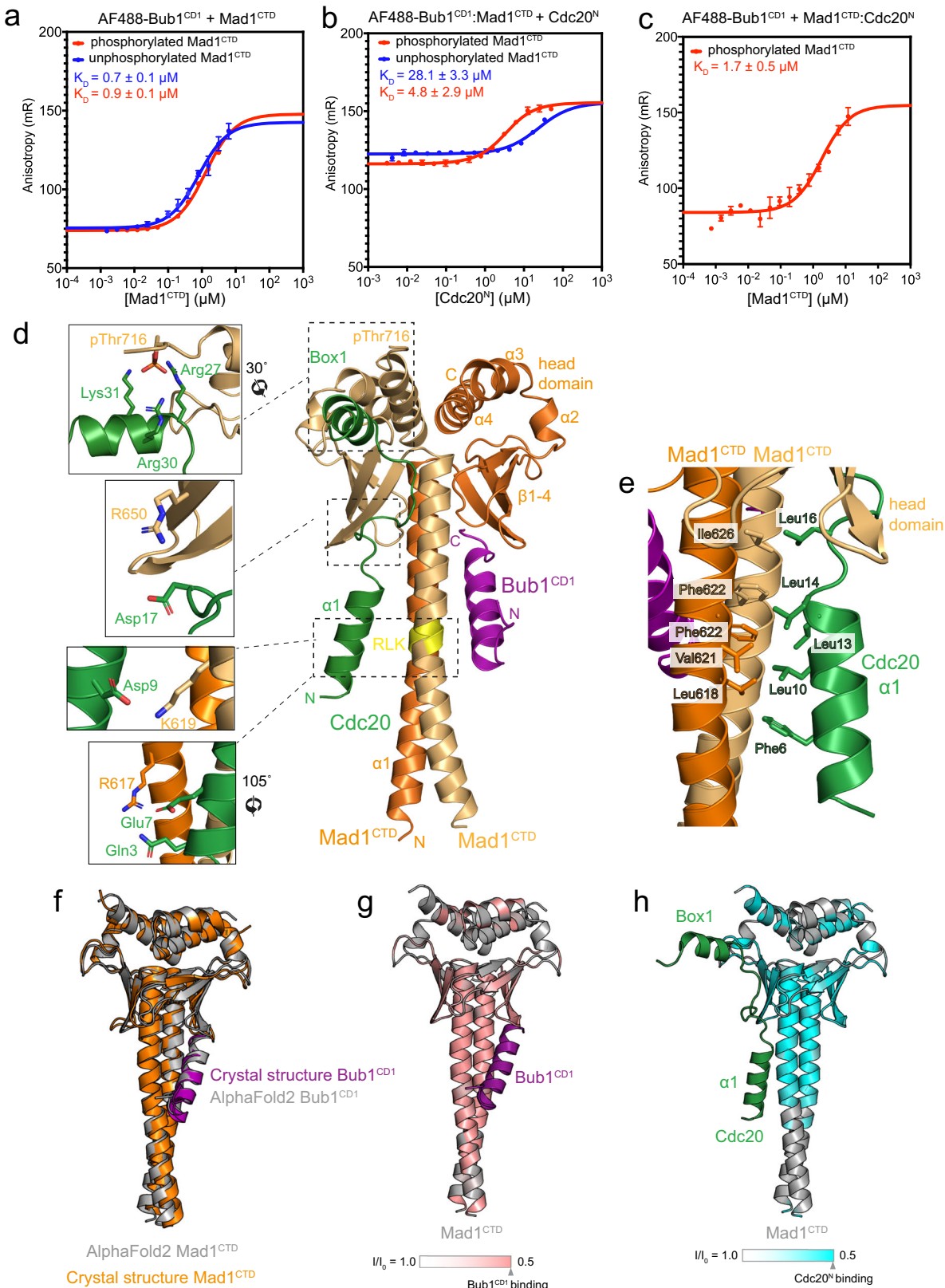

Cdc20[N] E7R/D9K mutant to pMad1[CTD] and found that their interaction could no longer be observed at the concentrations used in our fluorescent anisotropy assay or by SEC-MALS (Fig. 5). Additionally, as judged by SEC-MALS, wild-type Cdc20[N] did not bind a Mad1[CTD] R617E/K619D mutant (Fig. 5d, e). Because the Mad1[CTD] R617E/K619D mutant disrupted the binding of Bub1[CD1], fluorescent anisotropy could not be

used to further test the interaction of Mad1[CTD] R617E/K619D with wild-type Cdc20[N]. SEC-MALS did not show an interaction between Mad1[CTD] R617E/K619D and Cdc20[N] E7R/D9K, suggesting that this interaction could not simply be restored by swapping the charges. As mentioned earlier, mutation of the conserved QYRL motif (Q648A/R650A) within Mad1[CTD] impaired binding to Cdc20[33]. This is explained in our model

**Fig. 4 | A tripartite assembly of Cdc20 and Bub1 on phosphorylated Mad1$^{CTD}$.**
**a** Fluorescent anisotropy measurements with 20 nM AF488-Bub1$^{CD1}$ and titrating Mad1$^{CTD}$. The binding of Mad1$^{CTD}$ and Mad1$^{CTD}$ pThr716 to AF488-Bub1$^{CD1}$ gave calculated $K_D$ of 0.9 and 0.7 μM, respectively. **b** Fluorescent anisotropy measurements with preformed AF488-Bub1$^{CD1}$:Mad1$^{CTD}$ (20 nM: 2.1 μM) where Mad1 was phosphorylated or unphosphorylated and Cdc20$^N$ was titrated. The binding of Cdc20$^N$ to the Mad1$^{CTD}$:AF488-Bub1$^{CD1}$ complex where Mad1$^{CTD}$ were unphosphorylated and phosphorylated gave calculated $K_D$ of 28.1 and 4.8 μM, respectively. **c** Fluorescence anisotropy measurements for the titration of AF488-Bub1$^{CD1}$ to preformed pMad1$^{CTD}$:Cdc20$^N$ (20 nM: 15 μM), giving a $K_D$ of 1.7 μM. For data in **a**–**c**, error bars are the derived SD from three independent measurements. **d** AlphaFold2 prediction of the Mad1$^{CTD}$ homodimer (dark/light orange) bound to Bub1$^{CD1}$ (purple) and Cdc20$^N$ (green). The model was generated using Cdc20$^{1-73}$:Bub1$^{448-550}$:Mad1$^{597-718}$ with a defined stoichiometry of 1:1:2. Only the first 40 residues of Cdc20 are shown, including the α1 and Box1 α-helices. Cdc20$^N$:Mad1$^{CTD}$ interactions discussed in the text are highlighted in the left panels, where key residues are displayed as sticks.

pThr716 is shown to visualise how the phosphate is positioned to contact the positively charged residues of Box1 (top panel). **e** The hydrophobic interface between Cdc20$^N$ and Mad1$^{CTD}$ in the AlphaFold2 prediction. **f** Superimposition of the Mad1$^{CTD}$:Bub1$^{CD1}$ crystal structure[21] and the Mad1$^{CTD}$:Bub1$^{CD1}$ AlphaFold2 model shows their similarity, despite the Bub1$^{CD1}$ input sequence not containing phosphorylated Ser459 and Thr461 on Bub1$^{CD1}$. **g** Line broadening induced by Bub1$^{CD1}$ binding to $^{15}$N-labelled Mad1$^{CTD}$ pT716, as previously reported[21], was mapped onto the Mad1$^{CTD}$ AlphaFold2 model. Pink in the colour scale indicates regions with a significant line broadening and are likely involved in binding Bub1$^{CD1}$. The AlphaFold2 model of Bub1$^{CD1}$ is independently coloured in purple. **h** Line broadening induced by Cdc20$^N$ binding to Mad1$^{CTD}$ pT716, as reported in this study (Fig. 2d, e, g), was mapped onto the Mad1$^{CTD}$ AlphaFold2 model. Cyan in the colour scale indicates regions with a significant line broadening and are likely involved in binding Cdc20$^N$. The AlphaFold2 model of Cdc20$^N$ is independently coloured in green.

where Arg650 of Mad1$^{CTD}$ is predicted to contact Asp17 of Cdc20$^N$ (Fig. 4d).

In agreement with our NMR data (Fig. 3 and Supplementary Figs. 3, 5, 6), the Box1 motif α-helix, is predicted to bind the head domain near the C-terminus of Mad1$^{CTD}$. This positions three positively charged residues (Arg27, Arg30, Lys31) of Box1 near the negatively charged pThr716 of Mad1$^{CTD}$ (Fig. 4d), and this is consistent with our observation that Box1 interacts with pThr716 and the β2/β3 loop adjacent to pThr716 (Fig. 2g and Supplementary Fig. 6b, c). The AlphaFold2 prediction was generated without including a phosphorylated Thr716 in the input sequence. We assume that Thr716 of Mad1 co-evolves with the basic residues of Box1, and the multi-sequence alignment (MSA) algorithm of AlphaFold2 likely creates a close distance constraint between these residues[38]. This is supported by the accurate prediction of the Bub1$^{CD1}$:Mad1$^{CTD}$ interaction despite the sequence of Bub1$^{CD1}$ not containing a phosphorylated Thr461 which is essential for the Bub1$^{CD1}$:Mad1$^{CTD}$ interaction[14,20,21]. Mutation of Arg27, Arg30, Lys31 within Cdc20$^N$ to Ser (Cdc20$^N$ RRK/SSS), abolished its interaction with pMad1$^{CTD}$ as measured by SEC-MALS (Fig. 5d, e), and reduced the $K_D$ of their interaction as measured by fluorescent anisotropy to 23 μM (Fig. 5b, c). The comparable affinities of Cdc20$^N$ RRK/SSS binding for pMad1$^{CTD}$ and wild-type Cdc20$^N$ for unphosphorylated Mad1$^{CTD}$ (28 μM) is consistent with the interaction of these residues at the pThr716 site. This is supported by the similar affinity of Cdc20$^N$ RRK/SSS for unphosphorylated Mad1$^{CTD}$ (18 μM) (Fig. 5b, c). Altogether, the Cdc20$^N$:Mad1$^{CTD}$ AlphaFold2 prediction is in good agreement with our NMR data, illustrated by mapping the line broadening observed in the Cdc20$^N$:Mad1$^{CTD}$ titrations onto the model (Fig. 4h). It also explains how Cdc20 and Bub1 bind to Mad1$^{CTD}$ in a non-competitive manner to form a tripartite complex, as indicated by our fluorescence anisotropy assays (Fig. 4b), and the role of pThr716 in enhancing Cdc20$^N$ binding.

## Fold-over of Mad1$^{CTD}$ within the Mad1-Mad2 complex

One factor likely contributing to reducing the kinetic barrier for the formation of the C-Mad2:Cdc20 complex is a mechanism to promote the accessibility and proximity of Cdc20$^{MIM}$ relative to its binding site on Mad2[33,35]. Because Mad1 phosphorylation is required for catalysing MCC formation but not Cdc20 kinetochore recruitment[14,33,35], we next sought to further investigate how the tripartite assembly of Bub1$^{CD1}$, Cdc20$^N$ and Mad1$^{CTD}$ might contribute to catalysing the formation of the C-Mad2:Cdc20 complex. Mad1:C-Mad2, by means of O-Mad2 dimerisation to C-Mad2 and the interaction of Cdc20 to Mad1:Bub1, is the platform by which Mad2 and Cdc20 are brought together (Fig. 1b). Similar to other reports, we were unable to reconstitute a stable complex comprising the entire 'pre-MCC assembly' (pBub1:Bub3:p-Mad1:C-Mad2:O-Mad2:Cdc20)[12,14,33]. We, therefore, employed a

combination of cryo-EM, cross-linking mass spectrometry and SAXS to investigate the architecture of the Mad1:C-Mad2 complex (Fig. 1d).

We prepared Mad1:C-Mad2 tetramer and Mad1:C-Mad2:O-Mad2 hexamer, using Mad1 residues 485–718 (Mad1$^{485}$), in the phosphorylated and unphosphorylated states, and confirmed their homogeneity by SEC-MALS (Supplementary Fig. 8a, b). This truncated form of Mad1 is sufficient for catalytic MCC assembly in vitro[12]. The majority of phospho-peptides identified by mass spectrometry for Mps1-phosphorylated Mad1$^{485}$ contained a phosphorylated Thr716 (pThr716) (Supplementary Table 1). Lower abundant phospho-peptides with phosphorylation at Thr500, Ser538, Thr540, Thr550 and Ser551 were also detected (Supplementary Table 1). Intact mass spectrometry showed that each monomer of phosphorylated Mad1$^{485}$ contained predominantly only a single phosphorylated residue (Supplementary Fig. 1b), suggesting, and consistent with our previously presented data for Mad1$^{CTD}$, that Thr716 is the predominant Mps1 phosphorylation site within Mad1$^{485}$ (Supplementary Fig. 1a, c, d).

Two-dimensional class averages calculated from a cryo-EM data set of non-phosphorylated Mad1:C-Mad2:O-Mad2 revealed features corresponding to the Mad1:C-Mad2:O-Mad2 core and adjoining coiled-coil segments. The coiled-coil segment C-terminal to the Mad1:C-Mad2:O-Mad2 core is connected to Mad1$^{CTD}$ through a flexible linker region (Fig. 1d and Supplementary Fig. 9a–d) and matches the expected architecture of this complex (Supplementary Fig. 9e). These cryo-EM data allowed a 3D reconstruction of a medium resolution (10 Å) structure of the complex, although EM volumes for the flexible Mad1$^{CTD}$ were not recovered (Supplementary Fig. 9f, h). In contrast, 2D class averages of the phosphorylated Mad1:C-Mad2:O-Mad2 complex showed that conformational variability within the flexible linker region allowed this region to act as a hinge, with Mad1$^{CTD}$ able to adopt multiple conformations, including positions close to, and contacting the Mad1:C-Mad2:O-Mad2 core, as well as more extended states (Fig. 6a). This suggested that phosphorylated Mad1:C-Mad2:O-Mad2 exists in variable conformations ranging from a closed folded state (Mad1$^{CTD}$ proximal to, or in contact with the core), to an elongated state with Mad1$^{CTD}$ being distal to the core. Folded states of Mad1:Mad2 would be consistent with low-angle metal shadowing electron micrographs of Mad1:Mad2[33]. More recently, fold-over of Mad1 has also been detected in vivo using fluorescence-lifetime imaging (FLIM)[39]. The small size and conformational variability of this complex, despite been cross-linked, precluded a high-resolution cryo-EM reconstruction and suggests there is no stable folded state of Mad1$^{CTD}$ but that Mad1$^{CTD}$ remains highly dynamic (Supplementary Fig. 10a, b). A medium-resolution 3D reconstruction (11 Å at FSC = 0.143 or 16 Å at FSC = 0.5) of a folded state of Mad1 allowed the fitting of crystal structures[21,23,24,26] and visualisation of Mad1$^{CTD}$ positioned next to the core (Supplementary Fig. 10d, e, g). The presence of 'empty' density around Mad1$^{CTD}$ in this reconstruction is likely from this subset of particles containing Mad1$^{CTD}$ in

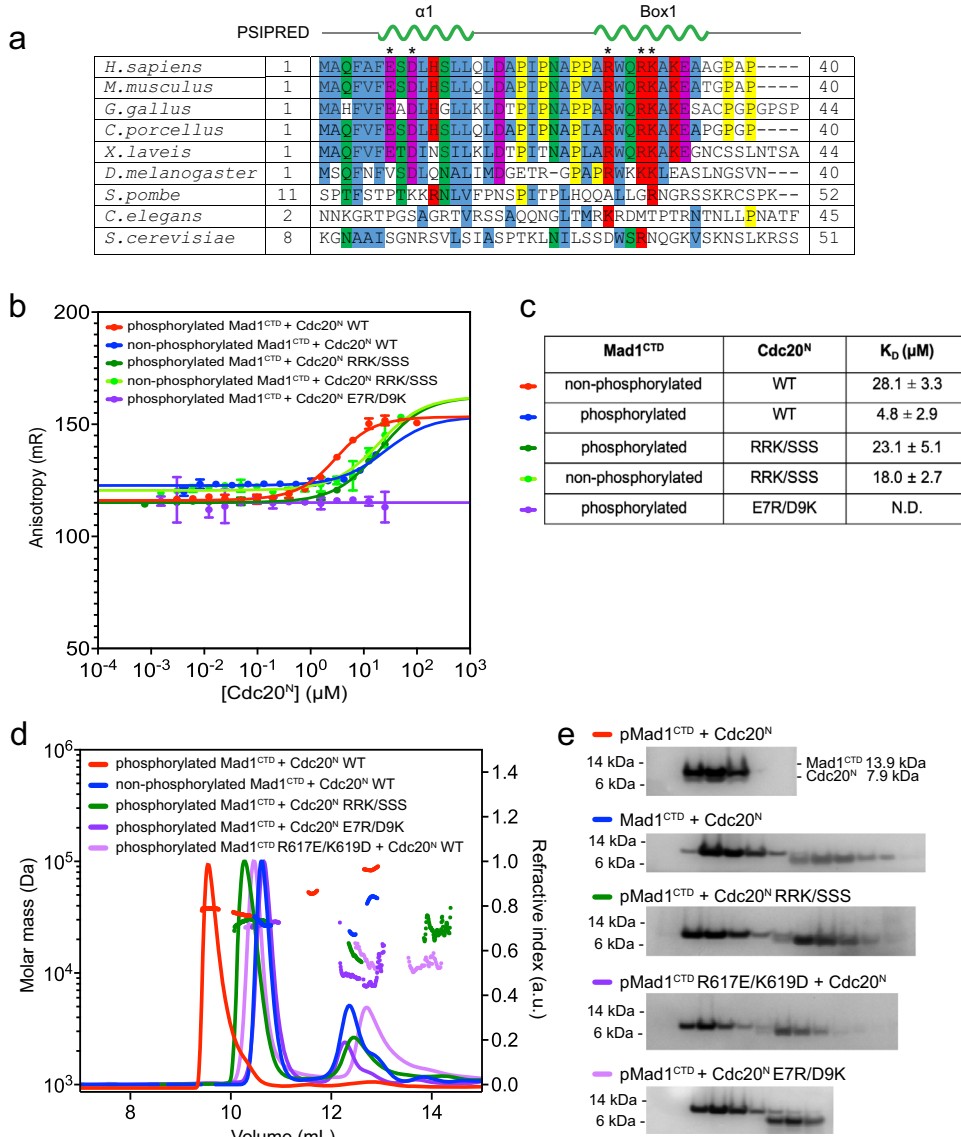

**Fig. 5 | Binding of Cdc20$^N$ and Mad1$^{CTD}$ mutants by fluorescent anisotropy and SEC-MALS. a** A multiple sequence alignment of Cdc20$^{1-40}$. Secondary structure prediction by PSIPRED is shown above for the α1 and Box1 predicted helices. The asterisks denote residues which were mutated and tested by fluorescent anisotropy or SEC-MALS in **b–d**. **b** Fluorescent anisotropy with preformed AF488-Bub1$^{CD1}$:Mad1$^{CTD}$ complex (20 nM: 2.1 μM) using pMad1$^{CTD}$ or Mad1$^{CTD}$ and where various Cdc20$^N$ mutants were titrated. The data were presented as mean values ± SD derived from three independent measurements. **c** Summary of the calculated $K_D$

of the interactions analysed in (**b**). **d** SEC-MALS analysis of Cdc20$^N$ interaction with pMad1$^{CTD}$ or Mad1$^{CTD}$ using various mutants. Theoretical masses for Mad1$^{CTD}$ dimer, Cdc20$^N$, and Mad1$^{CTD}$ dimer with a single Cdc20$^N$ bound are 28, 7.8 and 36 kDa respectively. **e** SDS-PAGE analysis of the SEC-MALS experiments shown in (**d**), using a 4–12% Bis-Tris Glycine gel with the SeeBlue™ Plus2 Protein Ladder (Thermo Fisher Scientific). Three independent measurements of each sample were completed with similar results.

various orientations. Density for the N-terminal helices was not recovered, which may be due to how the folded particles adhere to the EM grid, or due to Mad1$^{CTD}$ fold-over increasing their flexibility. Interestingly, AlphaFold2 predicted multiple conformations of Mad1$^{485}$ that varied between elongated and folded states depending on the degree of bending at the flexible linker region connecting the Mad1:C-Mad2:O-Mad2 core with Mad1$^{CTD}$ (Supplementary Fig. 8c–e). Modelling Mad1:C-Mad2:O-Mad2, based on structures of Mad1:C-Mad2 (PDB: 1GO4)[23] and the C-Mad2:O-Mad2 dimer (PDB: 2V64)[26], onto Mad1$^{MIM}$ in a folded state predicted by AlphaFold2 to have the highest confidence, generated a structure that resembles our cryo-EM reconstruction of a folded Mad1:C-Mad2:O-Mad2 state (Supplementary Fig. 10e and Fig. 6b).

To further investigate the conformation of Mad1$^{CTD}$ in hexameric Mad1:C-Mad2:O-Mad2 (phosphorylated and unphosphorylated) and tetrameric Mad1:C-Mad2 (unphosphorylated) states, we performed

in-gel cross-linking mass spectrometry (IGX-MS)[40] (Fig. 6b, c and Supplementary Fig. 11). We observed multiple cross-links between Mad1$^{CTD}$ and both Mad2 and the Mad1 N-terminal coiled-coils in all samples (Fig. 6b, c and Supplementary Fig. 11e–g). Mapping these cross-links onto the AlphaFold2-predicted Mad1:Mad2 structures indicated that the majority of cross-links are compatible with a folded conformation, but not with the elongated state (Fig. 6b, c). To further determine whether the cross-linking results distinguish between the elongated and folded states, a contact map for the respective Mad1$^{CTD}$ and Mad2 interfaces was generated (Fig. 6d). For simplicity, we calculated the distances only between Mad1$^{CTD}$ and the closest C-Mad2:O-Mad2 dimer. This analysis confirmed that the identified cross-links are consistent with a folded model of Mad1:C-Mad2:O-Mad2, and incompatible with the elongated conformation. Because the cross-linking pattern was similar for all three Mad1:Mad2 samples,

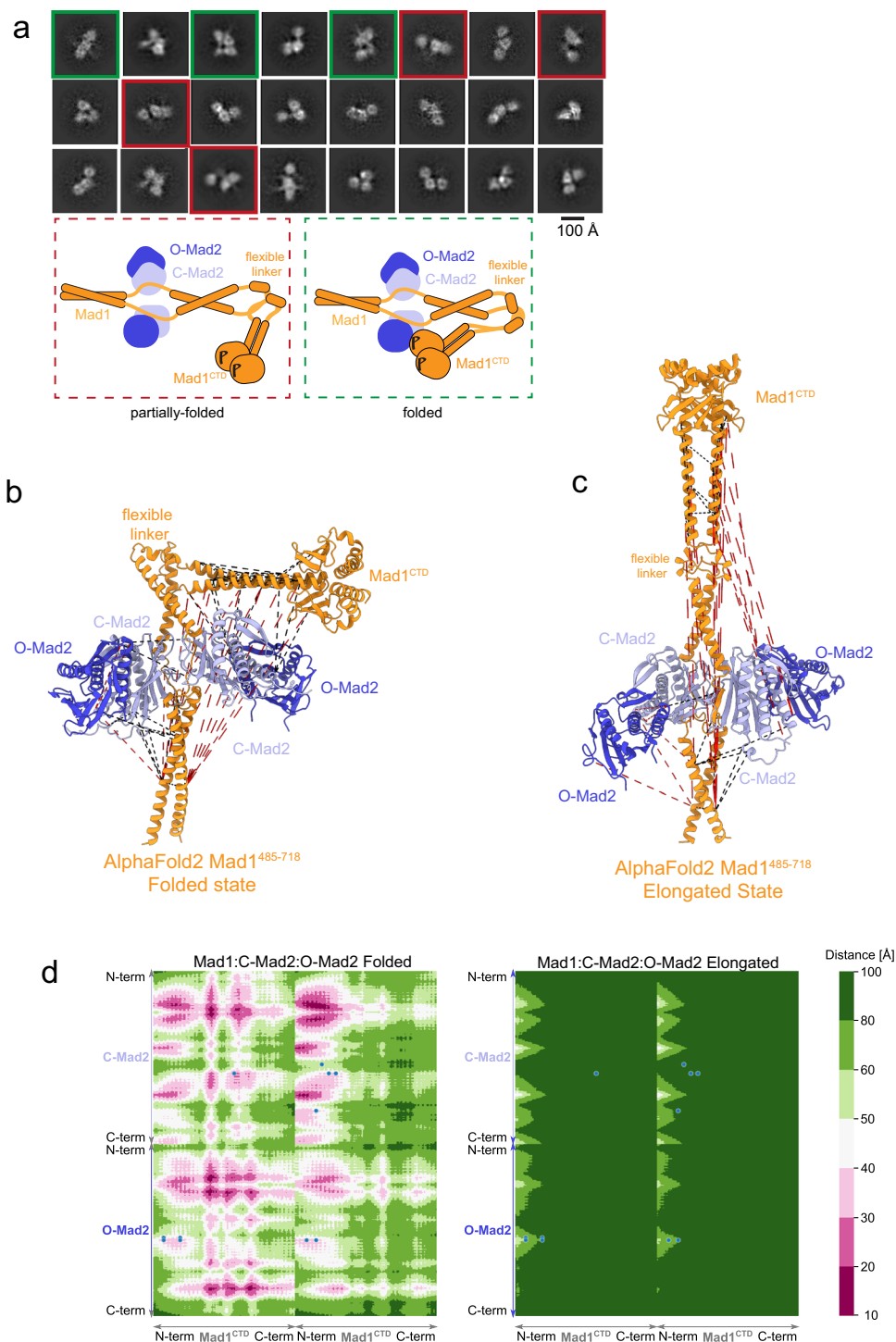

**Fig. 6 | Fold-over within the Mad1:Mad2 complex. a** 2D averages of BS3 cross-linked pMad1:C-Mad2:O-Mad2 complex by cryo-EM, representative of 125 2D classes. Classes showing complexes that appear only partially folded are boxed in red, and classes showing complexes that appear fully folded are boxed in green. A schematic of the suggested partially folded and folded states seen in the 2D averages is shown below. **b** Identified cross-links for the phosphorylated Mad1:C-Mad2:O-Mad2 complex were mapped onto a folded Mad1^CTD state as predicted by AlphaFold2 (Supplementary Fig. 8d) where alignment of the folded AlphaFold2 Mad1^485–718 structure onto the crystal structure of Mad1^485–584:C-Mad2 (PDB 1GO4)[23] and C-Mad2:O-Mad2 dimer (PDB 2V64)[26] was used to generate a model of the hexameric complex. Mad1 subunits (residues 485–718) are shown in orange,

whereas C-Mad2 and O-Mad2 are coloured in light and dark blue, respectively. Mapped cross-links with a distance below 40 Å are indicated as black dashed lines and all others in red. For all obtained cross-links, only the shortest option is depicted. **c** The same as in (**b**) but identified cross-links are mapped onto the structure of Mad1:C-Mad2:O-Mad2 in the elongated state as predicted by Alpha-Fold2 (Supplementary Fig. 8c). **d** Cross-link contact map for the folded and elongated Mad1:C-Mad2:O-Mad2 complexes. Cross-link sites reflecting inter Mad1^CTD:Mad2 cross-links are shown as blue circles. For simplicity, only contacts between dimeric Mad1^CTD and the closest Mad2 heterodimer (C-Mad2 and O-Mad2) were calculated. Intra- and intermolecular distances are shown in accordance with the coloured scale bar.

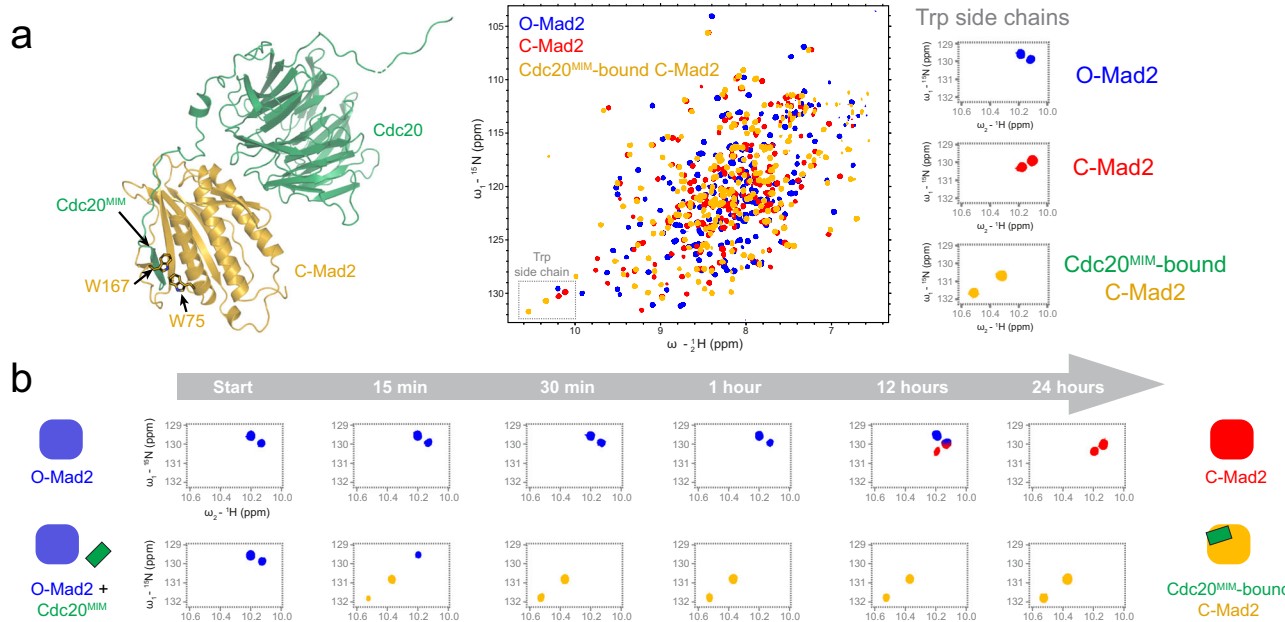

**Fig. 7 | NMR-monitor conversion of O-Mad2 to C-Mad2 shows that Cdc20[MIM] greatly enhances the rate of Mad2 conversion. a** C-Mad2 (orange) bound to Cdc20 (green) from PDB 6TLJ[90], with Mad2 Trp167 and Trp75 side chains depicted as sticks. The middle panel shows a ¹H, ¹⁵N 2D HSQC of ¹⁵N-labelled O-Mad2 (blue), C-Mad2 (red), and Cdc20[MIM]-bound C-Mad2 (orange). All Mad2 samples contain an R133A mutation to prevent Mad2 dimerisation[23]. The C-Mad2 sample additionally contains an L13A mutation which favours the C-Mad2 state[57]. Cdc20[MIM]-bound C-Mad2 sample was prepared by adding a 1:2 excess of Cdc20[MIM] peptide to O-Mad2 with an R133A mutation. The indole N-H resonances for Mad2 tryptophan side chains were used as reporters for Mad2 conformation and are boxed in the spectra. The right panel shows the close-up views of the two tryptophan indole N-H

resonances of O-Mad2 (blue), empty C-Mad2 (red), and Cdc20[MIM]-bound C-Mad2 (orange). As Cdc20[MIM] is entrapped by the safety-belt, the changes in the chemical environment of Trp167 and Trp75 result in distinct chemical shifts of their side chain resonances. **b** The MIM motif of Cdc20 induces the conversion of O-Mad2 into C-Mad2. Using the side chain resonances of Trp167 and Trp75 as reporters, the O-to-C conversion of Mad2 was traced in a 24-h time course at 25 °C. The schematics show the conformation of Mad2 as indicated by the tryptophan side chain resonances and the full spectra are shown in Supplementary Fig. 13a, b. O-Mad2 in the presence of Cdc20·MIM converts to Cdc20[MIM]-bound C-Mad2 (orange) within 30 min, while O-Mad2 alone takes up to 24 h to fully convert to C-Mad2 (red).

we conclude that both the phosphorylated Mad1:Mad2 hexamer and the unphosphorylated Mad1:Mad2 hexamer and tetramer are capable of adopting a folded state. However, we note that these data do not indicate the relative proportions of folded and elongated conformations at equilibrium, but rather highlight the high flexibility of Mad1[CTD]. In agreement with a compact model, cross-linking captures Mad1[CTD] orientations in close proximity (up to 30 Å) to C-Mad2:O-Mad2. Additionally, cross-linking also likely traps a more compacted state than is presented in the AlphaFold2 and cryo-EM models, explaining why even in our folded model, several of the cross-links were still identified as outliers (Fig. 6b and Supplementary Figs 8d, 10e).

Our IGX-MS experiments revealed that a folded state of the Mad1:Mad2 hexamer does not require Mad1[CTD] phosphorylation, contrasting with our cryo-EM results where a folded state was observed only for phosphorylated Mad1:Mad2. To assess the conformation and flexibility of the Mad1:Mad2 complexes in solution, we used size-exclusion-coupled small angle X-ray scattering (SEC-SAXS) (Supplementary Fig. 12). Our SEC-SAXS data showed that all Mad1:Mad2 complexes eluted as a single peak having a high degree of compositional homogeneity, with an average Rg across the peak of 46 Å. This Rg value compares to a calculated Rg of 43 Å for the folded states of the hexameric and tetrameric Mad1:Mad2 species, and 57 Å for the elongated states[41,42]. Plotting the pairwise distances between scattering points within the molecule, the P(r) distribution (Supplementary Fig. 12f), indicates that the maximum distance extends to approximately 176 Å ($D_{max}$) in all samples, suggesting there are no significant differences in the extent of Mad1[CTD] fold-over between the complexes. The fully extended Mad1:Mad2 model (Fig. 6c) has a maximum distance of ~202 Å, whereas the folded model (Fig. 5b) has a maximum

distance of approximately 151 Å. In addition, the hydrodynamic radius (Rh) was measured from the quasi-elastic light scattering (QELS) data collected during the SEC-MALS experiments (Supplementary Fig. 8a) giving a value of 53 Å (Supplementary Fig. 12g). The ratio of Rg/Rh is indicative of the overall shape of the complex[43]. The ratio of Rg collected from SEC-SAXS (46 Å) to the Rh (53 Å) from QELS of 0.87 compared to a value of 0.78 for a spherical mass, indicates a compact conformation. Thus, SEC-SAXS and SEC-MALS-QELS aligns with our IGX-MS experiments, allowing us to conclude that in solution, Mad1:Mad2 adopts variable conformations with a tendency to adopt a folded state, regardless of Mad1[CTD] Thr716 phosphorylation.

A possible explanation for why a folded state of non-phosphorylated Mad1:Mad2 was absent in our cryo-EM micrographs, is the marked preferred orientation of this molecule on cryo-EM grids (Supplementary Fig. 9g), in contrast to the random orientations of phosphorylated Mad1:Mad2 (Supplementary Fig. 10f). This indicates interactions of non-phosphorylated Mad1:Mad2 with the air-water interface that are likely to differentially affect the stability of the folded and elongated conformations.

### Cdc20[MIM] induces O-Mad2 to C-Mad2 conversion

We also used NMR to analyse the rate of O-Mad2 to C-Mad2 conversion (Fig. 7a, b and Supplementary Fig. 13). The substantial remodelling of O-Mad2 into C-Mad2 results in distinctly different ¹H, ¹⁵N HSQC spectra for Mad2 in different conformations (Fig. 7a). In particular insertion of Cdc20[MIM] near Trp75 and Trp167 in Cdc20[MIM]-bound C-Mad2 results in a marked chemical shift perturbation of their corresponding resonances (Fig. 7a)[27,44]. These resonances are useful reporters of Mad2 conformation in solution. We found that at 25 °C, O-Mad2 (100 µM) converted to C-Mad2 in 24 h (in line with previous reports)[12,29,32]

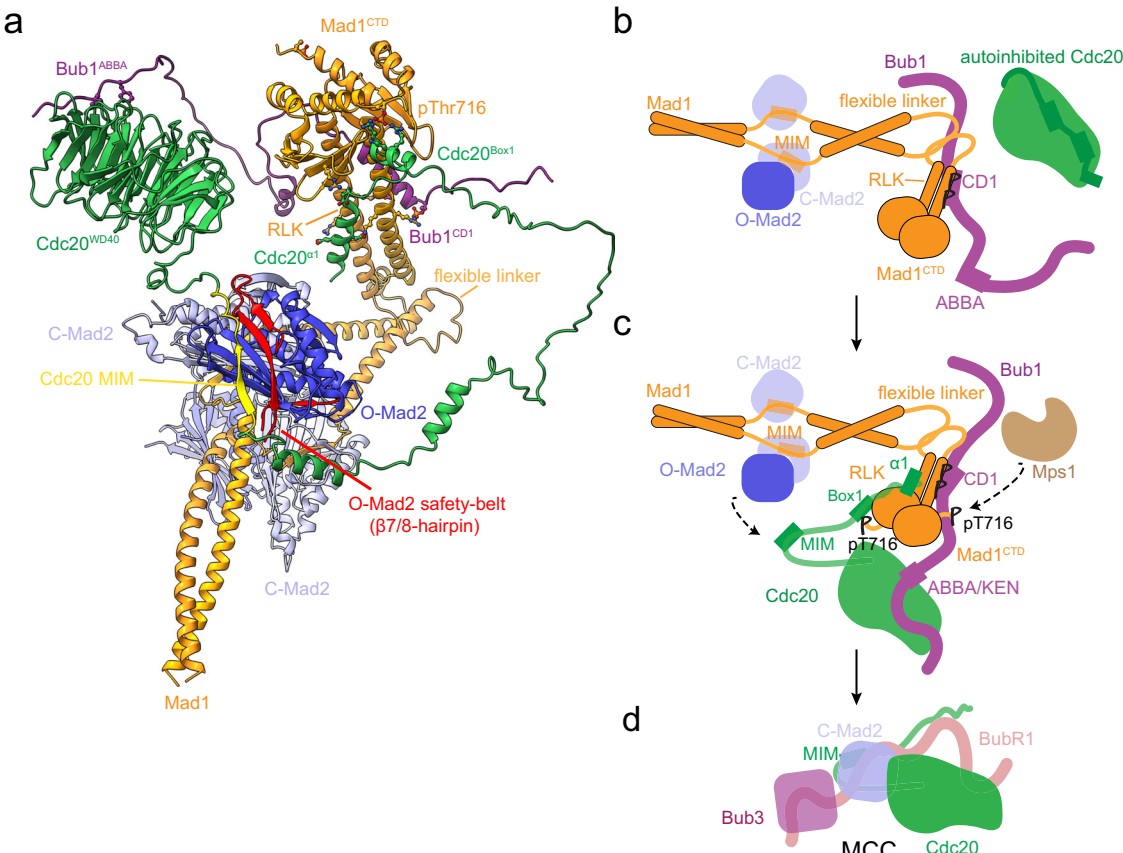

**Fig. 8 | A model for how tripartite assembly of pBub1:Cdc20:pMad1 in conjunction with Mad1^CTD fold-over orchestrates MCC formation. a** Model of pBub1:Cdc20:pMad1:C-Mad2:O-Mad2 complex (MCC-assembly scaffold) posed for MCC formation (Supplementary Movie 1). The model (Supplementary Data 1) was generated using PDBs 1GO4, 2V64, 6TLJ, 7B1F[21,23,26,90] and the AlphaFold2 models of Bub1^448–534:Mad1^CTD:Cdc20 and the folded model of Mad1^485–718, and displayed using ChimeraX. Residues for the Mad1 pThr716 and Cdc20^Box1, Cdc20^α1 Asp9 and Mad1 Lys619, Mad1 R617 and Bub1 pThr461 interactions are depicted as sticks, as well as the Bub1 ABBA interaction with the Cdc20 WD40 domain. **b–d** A schematic of catalytic MCC formation. **b** The doubly phosphorylated Bub1 CD1 domain targets the Mad1:C-Mad2 complex to kinetochores which then acts as a platform for O-Mad2 binding and O-to-C Mad2 conversion. Cdc20, on its own, likely exists in an autoinhibited state, which would impair the Cdc20:C-Mad2 interaction and MCC formation. **c** Phosphorylation of the C-terminus of Mad1 at Thr716 promotes its interaction with the N-terminus of Cdc20. Interaction between the WD40 domain of Cdc20 and the ABBA/KEN1 motif of Bub1 also occurs and this likely promotes Cdc20 kinetochore targeting and positions Cdc20 close to Mad1:C-Mad2. The Cdc20:pMad1^CTD interaction together with Mad1^CTD fold-over then promotes Cdc20-MIM accessibility and leads to Cdc20:C-Mad2 formation. **d** The Cdc20:C-Mad2 complex rapidly binds BubR1:Bub3 to form the MCC.

(Fig. 7b and Supplementary Fig. 13b). In the presence of only a two-molar excess of a Cdc20^MIM peptide (200 μM), complete Mad2 conversion and entrapment of Cdc20^MIM occurred in less than 30 min (Fig. 7b and Supplementary Fig. 13a). We also showed that such spontaneous conversion cannot be achieved using a substoichiometric concentration of Cdc20^MIM (1:50 to Mad2, Supplementary Fig. 13c). This suggests that Cdc20^MIM functions trigger a conversion and stabilises Cdc20^MIM-bound C-Mad2. The capacity of Cdc20^MIM to induce the O-Mad2 to C-Mad2 conversion is reminiscent of the O-Mad2 to C-Mad2 conversion induced by Mad1^MIM [ref. 29].

## Discussion

Phosphorylation of Mad1^CTD by Mps1, that functions to promote interaction with Cdc20^N, together with Mps1-mediated phosphorylation of Bub1, have been identified as crucial steps in catalysing MCC assembly[14,33,35]. Additionally, O-Mad2 dimerisation with C-Mad2 contributes to catalysing the O-Mad2 to C-Mad2 conversion, although at a more modest rate that does not account for the rate of MCC assembly in cells during an active SAC[13,32]. Here, we found that the Mps1 phosphorylation of Mad1 Thr716 directly promotes an extensive interaction between Cdc20^N and Mad1^CTD. We find that, similar to Bub1^CD1, only one Cdc20^N binds to the Mad1^CTD homodimer. Cdc20 and Bub1 share a comparable binding interface on Mad1^CTD, but instead of competing,

the two proteins bind simultaneously to Mad1^CTD, forming a tripartite complex. We employed AlphaFold2 to predict a model of Bub1^CD1:Cdc20^N:Mad1^CTD. The resultant model was consistent with our NMR and biophysical characterisation of the Cdc20^N:Mad1^CTD interaction, and correctly predicted our previously reported Bub1^CD1-Mad1^CTD crystal structure[21]. It also illustrated how dimeric Mad1^CTD can accommodate both Bub1^CD1 and Cdc20^N. Using cryo-EM and IGX-MS, we additionally identified a mechanism by which Mad1^CTD folds over towards Mad1-bound C-Mad2.

Utilising our discovery of a tripartite Bub1^CD1:Cdc20^N:Mad1^CTD complex, together with Mad1^CTD fold-over, we can present a plausible model for the structure of the catalytic scaffold posed for MCC formation (Fig. 8a and Supplementary Movie 1). Importantly, this model outlines how this assembly would position Cdc20^MIM in close proximity to O-Mad2 for their interaction (Fig. 8a–d). Detachment of the safety-belt (β7/8 hairpin) concomitant with the O-Mad2 to C-Mad2 conversion would trap Cdc20^MIM within C-Mad2. In our model (Fig. 8a), the configuration of Cdc20 in the context of the MCC-assembly scaffold orientates Cdc20^MIM antiparallel to the β6 strand of O-Mad2, with which it pairs as an antiparallel β-sheet in the C-Mad2:Cdc20 complex. Thus, both the close proximity and optimal orientation of Cdc20^MIM relative to O-Mad2, facilitated by the MCC-assembly scaffold, would contribute to catalysing C-Mad2:Cdc20 formation.

This model also demonstrates how the tripartite assembly of Bub1[CD1] and Cdc20[N] on Mad1[CTD] is compatible with Bub1[ABBA]-mediated Cdc20 kinetochore recruitment[45–48]. The parallel orientation of Bub1[CD1] and Cdc20[N] with respect to Mad1[CTD], optimally positions Bub1[ABBA], located C-terminal to Bub1[CD1], and the C-terminal WD40 domain of Cdc20 (Cdc20[WD40]) for their interaction (Fig. 8a, c). Self-interaction of the N- and C-termini of Cdc20 has been reported[35,49,50], leading to the suggestion that the catalytic scaffold also acts to relieve Cdc20 autoinhibition[33,35]. Our model suggests dual anchorage of Cdc20[N] to pMad1[CTD] and Cdc20[WD40] to Bub1[ABBA] as a plausible mechanism for both relieving Cdc20 autoinhibition and promoting the accessibility of Cdc20[MIM] for Mad2 safety-belt entrapment. This also likely explains why both Bub1[CD1] and Bub1[ABBA] contribute to catalytic MCC assembly in vitro[33], and how a viable checkpoint requires Bub1[CD1] even when Mad1 is tethered to kinetochores[20,51,52].

Our model for the catalytic scaffold also provides insight into the relevance of Mad1[CTD] fold-over, mediated through the flexible linker. In the elongated state of Mad1 (Supplementary Fig. 14), the Mad1[CTD] is separated from the Mad1:Mad2 dimerisation platform compared to the Mad1 folded state (Fig. 8a). Modelling suggests that the segments connecting both Bub1[CD1] with Bub1[ABBA] and Cdc20[N] with Cdc20[MIM] are of insufficient length to allow Cdc20[MIM] to be positioned proximal to the O-Mad2 safety-belt. This is supported by a study which reported that catalytic MCC formation is impaired by decreasing the distance between the Cdc20 Box1 and MIM motifs[33]. In a more recent complementary study, Chen and colleagues demonstrate that the flexible hinge within Mad1 which allows Mad1[CTD] fold-over is essential for catalytic MCC assembly in vitro and SAC signalling in vivo[39]. They also identified that catalysis induced by the flexible linker was dependent upon an intact Mad1:Mad2 interaction with Bub1[CD1] and Cdc20[Box1], thereby coupling Mad1 flexibility and tripartite Bub1[CD1]:Cdc20[N]:Mad1[CTD] assembly with catalytic Mad2:Cdc20 formation.

Our observation that Cdc20[MIM] induces the O-Mad to C-Mad2 conversion is significant because it further confirms that the key function of the catalytic scaffold is to promote the accessibility and repositioning of Cdc20[MIM] for Mad2 binding. Ultimately, this likely explains how Mad1 phosphorylation and its interaction with Cdc20 accelerates MCC formation and provides significance to our model of Cdc20[MIM] poised to bind Mad2 and subsequently induce Mad2 conversion and, therefore its own incorporation into the MCC (Fig. 8a).

In summary, our study provides mechanistic insights into how Mps1 phosphorylation of Mad1 modulates interaction with the N-terminus of Cdc20, completing a catalytic scaffold which acts to promote the Cdc20:C-Mad2 interaction, thereby overcoming the key kinetic barrier to MCC formation. Understanding how Cdc20[MIM] directly induces the O-Mad2 to C-Mad2 conversion is an outstanding future challenge.

## Methods

### Expression and purification

A gallery of all purified proteins in this study by SDS-PAGE is shown in Supplementary Fig. 15. Mad1[CTD] was expressed and purified as previously described[21]. The coding region of human Cdc20[N] was cloned by USER® (NEB) into a modified pRSFDuet-1 vector (71341-3, Sigma-Aldrich) with an N-terminal His₆-MBP-tag, followed by a tobacco etch virus (TEV) protease cleavage site[53,54]. Cdc20 was transformed into Rosetta™ 2 (DE3) Singles™ Competent Cells (71400, Novagen) for expression. Expression was induced with 0.3 mM IPTG and grown overnight at 20 °C. Cells were lysed in 25 mM HEPES pH 8.2, 300 mM NaCl, 10% glycerol, supplemented with lysozyme, and Complete™ EDTA-free protease inhibitors (Roche). His₆-MBP-Cdc20[N] was then purified over a HiTrap TALON® Crude column (Cytiva), and the tag was cleaved with TEV protease overnight at 4 °C. Cleaved Cdc20[N] was then passed over a HisTrap column (GE Healthcare) and two rounds of cation-exchange Resource S (GE Healthcare) and Superdex 75 (GE

Healthcare) purification. The size-exclusion and storage buffer consisted of 20 mM HEPES pH 7.0, 100 mM NaCl, 1 mM TCEP. Mad1[CTD] and Cdc20[N] mutants were generated using the QuikChange™ Lightning Site-Directed Mutagenesis Kit (Agilent), developed by Stratagene Inc. (La Jolla, CA)[55].

The Mad1:C-Mad2 tetrameric complexes (His₆-DoubleStrep-TEV-Mad1[485–718]:Mad2 and His₆-DoubleStrep-TEV-Mad1[485–718]:Mad2[R133A]), as well as GST-Mps1, were cloned into a modified MultiBac system as previously described[56]. The Mad1:Mad2 complexes were purified as previously described[14]. Mad2 R133A is a dimerisation-deficient Mad2 mutant and was used when forming a tetrameric-only control complex to prevent super-stoichiometric amounts of Mad2 being present due to Mad2 self-dimerisation[25]. We found that during baculovirus expression, the amount of Mad2 in the tetrameric complex was sub-stoichiometric, and therefore we individually purified Mad2 WT or Mad2 R133A with an N-terminal double strep-tag followed by a TEV protease site using the same protocols as for Mad1[CTD] [ref. 21]. Excess Mad2 was added to the Mad1:C-Mad2 complex and re-purified by SEC to produce a homogenous 2:2 complex of Mad1:C-Mad2. Mad1:C-Mad2:O-Mad2 hexameric complex was formed by adding an excess of Mad2LL (Mad2 kinetically locked in the open state), using the same construct and purification strategy as previously described[26]. The homogeneity of the various Mad1:Mad2 tetrameric and hexameric complexes was confirmed by SEC-MALS (Supplementary Fig. 8a, b). Expression and purification of Mad2 in the open state for use in the Mad2 conversion NMR experiments was carried out as previously described in ref. 57.

GST-Mps1 was purified using a lysis buffer of 25 mM HEPES, pH 7.0, 500 mM NaCl, 10% glycerol and 2 mM DTT supplemented with 25 mg/ml lysozyme, and 0.05% NP-40 and purified over 10 mL of glutathione sepharose™ 4B resin. GST-Mps1 was eluted overnight at 4 °C without removal of the GST-tag to keep full-length Mps1 soluble, using an elution buffer containing 25 mM HEPES pH 8.0, 500 mM NaCl, 2 mM DTT, 10% glycerol and 20 mM reduced glutathione. The eluate was concentrated and then run over a Superdex 200 16/600 column in 20 mM HEPES pH 8.0, 100 mM NaCl and 1 mM TCEP.

### Peptide synthesis

Cdc20[MIM]: NLNGFDVEEAKILRLSGKPQNAPEGYQNRLKVLY, was synthesised by Designer Bioscience, UK. All other peptides were synthesised by Cambridge Research Biochemicals, UK. Cdc20 α1: MAQFAFESDLHSLL. Cdc20 Box1: PPARWQRKAKEAA. AF488-Bub1[CD1]: [Cys(AF(488)]-W-KVQP-[pSer]-P-[pThr]-VHTKEALGFIMNMFQAPTS, where AF488 is Aurora Fluor 488 and is structurally identical to Alexa Fluor®488 C5 maleimide.

### NMR spectroscopy

Uniformly-labelled proteins were expressed in M9 minimal media (6 g/L Na₂HPO₄, 3 g/L KH₂PO₄, 0.5 g/L NaCl) supplemented with 1.7 g/L yeast nitrogen base without NH₄Cl and amino acids (Sigma Y1251). 1 g/L ¹⁵NH₄Cl and 4 g/L unlabelled glucose were supplemented for ¹⁵N labelling. For ¹³C/ ¹⁵N double-labelled samples, unlabelled glucose was replaced with 3 g/L ¹³C-glucose. Prior to all NMR experiments, proteins were dialysed into 20 mM HEPES, pH 7.0, 100 mM NaCl and 1 mM TCEP.

The following protein concentrations were used in NMR studies: ¹H,¹⁵N HSQC of Mad1[CTD] were collected with a 100 μM sample, whereas a 50 μM sample was used for titrations with Cdc20[N] and Cdc20[α1]. Cdc20[Box1] peptide was added at mM concentrations to minimise dilution during titration. Limited by solubility, Cdc20[N] or Cdc20[α1] peptides were pre-mixed with Mad1[CTD] to achieve a final concentration of 50 μM Mad1[CTD]. ¹H,¹⁵N HSQC and ¹³C,¹⁵N CON of Cdc20[N] were collected at 50 and 200 μM respectively. Mad1[CTD] was concentrated to 3 mM for titrations into Cdc20[N]. ¹H,¹⁵N HSQC of Mad2 were collected at 100 μM. All Mad2 NMR samples contain an R133A mutation to prevent Mad2 dimerisation[23]. The C-Mad2 sample additionally contains an L13A

mutation which favours the C-Mad2 state[58]. Cdc20$^{MIM}$-bound C-Mad2 sample was prepared by adding a 1:2 excess of Cdc20$^{MIM}$ peptide to an O-Mad2 sample with R133A mutation. To follow the O-to-C conversion of Mad2, the samples were prepared on ice and monitored using BEST $^1$H,$^{15}$N-TROSY HSQC. The spectra were acquired with 16 scans and a recycle delay of 400 msec, giving an experimental time of 12 min per spectrum.

Mad1$^{CTD}$ and Mad2 experiments were performed using an in-house 800 MHz Avance III spectrometer and Cdc20$^N$ experiments were performed using an in-house 700 MHz Avance II + spectrometer, both equipped with triple resonance TCI CryoProbes. To retrieve signals that suffer from exchange broadening, we also utilised the Bruker 950 MHz Avance III spectrometer located at MRC Biomedical NMR Centre (Francis Crick Institute). Mad1$^{CTD}$ and Mad2 spectra were collected at 298 K. Cdc20$^N$ spectra were collected at 278 K to minimise exchange broadening for the largely disordered Cdc20$^N$.

Backbone resonances of Cdc20$^N$ were obtained with BEST TROSY versions of triple resonance experiments: HNCO, HN(CA)CO, HNCACB and HN(CO)CACB (Bruker). All 3D datasets were collected with non-uniform sampling at 10–20% and processed in MddNMR[59] using compressed sensing reconstruction. Additional non-amide resonances were assigned using $^1$H start versions of $^{13}$C-detected CON, CBCACON and CBCANCO[60]. Backbone resonances were assigned using in-house scripts and Mars[61]. Topspin 4.1.1 (Bruker) was used for processing and NMRFAM-Sparky 1.47 for data analysis[62].

$^{15}$N{$^1$H}-heteronuclear NOE values were measured from technical duplicates using standard Bruker pulse sequences and expressed as I/I$_0$ ratio, with interleaved on- (I) and off-resonance (I$_0$) saturation, using a recycle delay of 5 seconds. Secondary chemical shifts were calculated using the equation δCa$_{obs}$ - δCa$_{rc}$ where δCa$_{obs}$ are the observed Cα chemical shifts and δCa$_{rc}$ are the Cα chemical shifts for random coils[36,37]. Random coil chemical shifts were obtained by assigning the backbone resonances of Cdc20$^N$ in the presence of 2 M urea using the experiments described above.

For binding studies, the relative peak intensities were normalised to the C-terminal residue Ala718 of Mad1$^{CTD}$ or Gly73 of Cdc20$^N$ and expressed as PI$_{bound}$/PI$_{free}$, with PI$_{free}$ and PI$_{bound}$ being the peak heights of the free and bound forms, respectively. Weighted chemical shift perturbations were calculated using the equation[63]: $\Delta\delta = [(\Delta\delta_{HN}W_{HN})^2 + (\Delta\delta_N W_N)^2)^2]^{1/2}$ where $\Delta\delta_{HN}$ and $\Delta\delta$N are the chemical shift perturbations in $^1$H and $^{15}$N dimensions respectively[64]. The weight factors were determined from the average variances of chemical shifts in the BMRB database[65], where W$_{HN}$ = 1 and W$_N$ = 0.16.

## In vitro protein phosphorylation

Mad1$^{CTD}$ or Mad1:Mad2 complexes and GST-Mps1 were buffer exchanged into 25 mM HEPES pH 8.0, 5% glycerol, 0.1 μM okadaic acid, 15 mM BGP, 2 mM ATP, 10 mM MgCl$_2$, 0.5 mM TCEP, using a PD-10 column (GE Healthcare). Mad1 and GST-Mps1 were mixed at a 5:1 ratio and incubated at 28 °C for 3.5 h before removing the phosphorylation buffer and GST-Mps1 from Mad1 by size-exclusion chromatography.

## Mass spectrometry

Identification of phosphorylation sites on Mad1$^{CTD}$ and Mad1:C-Mad2 was performed at the MRC-LMB mass spectrometry facility. Intact proteins were subjected to LC-MS analysis. A modified NanoAcquity (Waters, UK) delivered a flow of approximately 50 μl/min, and proteins were injected directly on a C4 BEH 1.7 μm, 1.0 × 100 mm UPLC column (Waters, UK). Proteins were eluted with a 20-min gradient of acetonitrile (2 to 80%). The analytical column outlet was directly interfaced via an electrospray ionisation source, with a hybrid quadrupole time-of-flight (Q-TOF) mass spectrometer (Xevo G2, Waters, UK). Data were acquired over an m/z range of 350–2000, in positive ion mode with a cone voltage of 30 v. Scans were summed together manually and deconvoluted using MaxEnt1 (Masslynx, Waters, UK).

Peptide mass spectrometry was completed after reduction with 10 mM DTT and alkylation with 55 mM iodoacetamide and protein digestion overnight at 37 °C at a 1:50 ratio of trypsin (Promega, UK). Tryptic peptides were analysed by nano-scale capillary LC-MS/MS with an Ultimate U3000 HPLC (ThermoFisher Scientific Dionex, San Jose, USA) set to a flow rate of 300 nL/min. Peptides were trapped on a C18 Acclaim PepMap100 5 μm, 100 μm × 20 mm nanoViper (Thermo Fisher Scientific Dionex, San Jose, USA) prior to separation on a C18 T3 1.8 μm, 75 μm × 250 mm nanoEase column (Waters, Manchester, UK). A gradient of acetonitrile eluted the peptides, and the analytical column outlet was directly interfaced using a nano-flow electrospray ionisation source, with a quadrupole Orbitrap mass spectrometer (Q-Exactive HFX, ThermoFisher Scientific, USA). For data-dependent analysis a resolution of 60,000 for the full MS spectrum was used, followed by twelve MS/MS. The data were then searched against an LMB in-house database using a Mascot search engine (Matrix Science)[66], and the peptide identifications were validated using the Scaffold programme (Proteome Software Inc.)[67].

## SEC-MALS

Size-exclusion chromatography coupled with multi-angle static light scattering (SEC-MALS), was performed using an Agilent 1200 series LC system with an online Dawn Helios ii system (Wyatt) equipped with a QELS + module (Wyatt) and an Optilab rEX differential refractive index detector (Wyatt). About 100 μl purified protein at ~3.0 mg/ml was auto-injected onto Superdex 75 or a Superdex 200 Increase 10/300 GL column (GE Healthcare) and run at 0.5 ml/min. The molecular masses were analysed with ASTRA 7.3.0.11 (Wyatt). Data were plotted using Prism 8.4.3 (GraphPad Software, Inc). Purified Cdc20$^N$ and Cdc20$^N$ mutants were incubated with Mad1$^{CTD}$ wild-type or mutants at a 1:1.2 ratio of Mad1$^{CTD}$ dimer to Cdc20$^N$.

## AUC-SE

Sedimentation equilibrium experiments were carried out in a buffer of 20 mM HEPES pH 7.5, 100 mM NaCl and 1 mM TCEP. Samples were loaded into 12 mm six-sector cells, placed in an An-50Ti rotor and centrifugated at 10,200, 12,200 and 21,000 rpm at 20 °C until equilibrium had been reached using an Optima XL-I analytical ultra-centrifuge (Beckman). The data were analysed in SEDPHAT 15.2b[68]. The partial-specific volumes (v-bar) were calculated using Sednterp[69]. The density and viscosity of the buffer were determined with a DMA 4500 M density metre (Anton Parr) and an AMVn viscometer (Anton Paar). Data were plotted with the program GUSSI[70].

## Fluorescent anisotropy

All reactions were completed in a buffer of 25 mM HEPES pH 7.75, 1 mM TCEP, 100 mM NaCl, 0.05% (v/v) Tween-20. To analyse the binding of Bub1$^{CD1}$ to Mad1$^{CTD}$, a twofold dilution series of Mad1$^{CTD}$ from 100 μM to 6 nM was prepared and mixed 1:1 with fluorescently-labelled AF488-Bub1$^{CD1}$ at 40 nM. To analyse the binding of Cdc20$^N$ to the Mad1$^{CTD}$: AF488-Bub1$^{CD1}$ complex, a mixture of 40 nM AF488-Bub1$^{CD1}$ and 4.2 μM Mad1$^{CTD}$ was added 1:1 with a twofold dilution series of Cdc20$^N$ from 100 μM to 6 nM. To analyse the binding of Bub1$^{CD1}$ to the Mad1$^{CTD}$:Cdc20$^N$ complex and test for cooperativity, a twofold dilution series of Mad1$^{CTD}$ from 96 μM to 2.9 nM was mixed 1:1 with 92 μM of Cdc20$^N$, incubated on ice for 30 min and then mixed 1:1 with 40 nM of AF488-Bub1$^{CD1}$. Measurements were performed with a PheraStar plate reader FSX plate reader (BMG Labtech) using an optic module for $\lambda_{ex}$ = 485 nm, $\lambda_{em}$ = 520 nm. Reactions were carried out in a total volume of 40 μl at 25 °C in a black, flat-bottom, non-binding surface 384-well plate (Corning). All experiments were performed as technical triplicates and data were analysed in PRISM 9.3.1 (GraphPad Software). For titration of wild-type Cdc20$^N$ and Cdc20$^N$ the fit for the top value of the anisotropy was constrained to be a shared value between the phosphorylated and non-phosphorylated Mad1$^{CTD}$-Bub1$^{CD1}$ samples, as

further titration of Cdc20$^N$ to saturation was not possible due to large increases in the total fluorescence intensity indicating aggregation or non-specific binding to the plate.

## BS3 cross-linking for cryo-EM

Purified homogenous Mad1:C-Mad2:O-Mad2 complex in the unphosphorylated or phosphorylated state were cross-linked with BS3 (bis[sulfosuccinimidyl] suberate) (Thermo Fisher Scientific) before analysis by cryo-electron microscopy. About 2 mM BS3 was incubated with Mad1:Mad2 complexes at 0.3 mg/ml for 1 h at 4 °C. BS3 was then quenched by the addition of Tris-HCl pH 8.0 to 100 mM and incubated on ice for 30 min. The cross-linked complex was then concentrated to 3.5 mg/ml and re-purified by size-exclusion chromatography, first on a Superdex 200 15/300 column (GE Healthcare), and then again on a Superdex 200 5/150 (GE Healthcare).

## Cryo-EM grid preparation and data acquisition

Copper Quantifoil 1.2/1.3, 300 mesh, grids were used. Grids were treated with a 9:1 argon: oxygen plasma for 45 s, after which 3.5 µl of the purified complex at 1.5–2.5 mg/ml supplemented with 0.05% NP-40 (1 h before grid application) was pipetted onto the grid at 4 °C and 100% humidity before immediately blotting for 2 s at blot force −10 and plunging into liquid ethane using an FEI Vitrobot Mark III (Thermo Fisher Scientific). Data were collected at 300 kV on a Titan Krios, equipped with a K3 detector with a pixel size of 1.1 Å (Gatan). A volta-phase plate (VPP) was additionally used with the phosphorylated Mad1:C-Mad2:O-Mad2, as the VPP was found to improve the data quality and allowed seeing the coiled-coil connection between the flexible Mad1$^{CTD}$ and core. Data acquisition statistics are outlined in Supplementary Table 2.

## Cryo-EM data processing

Data were pre-processed using Warp v1.09, including training and then implementing a model for particle picking[71]. All further processing was carried out in CryoSparc v2.15.0[72]. Particles were cleaned up using repetitive 2D classifications. Initial 3D reconstructions were created using an SGD algorithm[72]. For the unphosphorylated Mad1:C-Mad2:O-Mad2 complex, heterogenous refinement was used to further clean up the data, after which homogenous refinement was used to obtain a final resolution of 8.9 Å (Supplementary Table 2). For the phosphorylated Mad1:C-Mad2:O-Mad2 complex, 2D classification was used to remove open-state particles (7.6% of the total particles) and particles where the head domain was not closely folded towards the core (12.3% of total particles). Further repetitive rounds of 2D classification were performed until classes which clearly presented the head domain folded towards the core with the coiled-coil being visible. These classes were then used to make an initial model (4,714 particles). Attempts at 3D classification and heterogenous refinement did not help to produce a higher resolution map, due to the high flexibility of this complex. Many different models with the head domain in slightly different degrees of folding could be obtained, and no stable conformation could be identified. Ultimately non-uniform refinement in cryoSPARC was used to produce a model for a folded state of Mad1:Mad2 with a final resolution of 11.1 Å at 0.143 FSC and 16.3 Å at 0.5 FSC[73]. The reported estimated final map resolutions were calculated based on the gold standard Fourier shell correlation (GS-FSC) 0.143 or 0.5 criterion[74–77]. Crystal structures of Mad1$^{CTD}$ (PDB ID: 4DZO[24]), Mad1:C-Mad2 (PDB ID: 1GO4[23]) and C-Mad2:O-Mad2 (PDB ID: 2V64[26]) were manually docked into the EM density maps using in Chimera[78].

## In-gel cross-linking mass spectrometry (IGX-MS)

Purified complexes of the tetrameric Mad1$^{485–718}$:C-Mad2 (using Mad2$^{R133A}$) complex, as well as the unphosphorylated and phosphorylated hexameric Mad1$^{485–718}$:C-Mad2:O-Mad2 complexes (with O-Mad2 being the Mad2LL mutant described earlier) were cross-linked

following the published IGX-MS workflow[40]. Respective complexes were mixed with NativePAGE sample buffer, and subsequently, 15 µg of each sample was run onto a Bis-Tris gel (3–12%). For cross-linking, bands corresponding to tetrameric and hexameric Mad1:Mad2 complexes were excised, rinsed with distilled H$_2$O and subsequently cross-linked (in triplicates) with 1.5 mM DSS for 30 min at 25 °C. The reaction was quenched by the addition of 1 M Tris pH 8.5 to a final concentration of 50 mM. Following classical in-gel digestion[79], cross-linked proteins were washed, reduced and alkylated prior to trypsin digestion. For MS analysis of cross-linked peptides, the samples were resuspended in 2% formic acid and analysed using an UltiMate™ 3000 RSLCnano System (Thermo Fisher Scientific) coupled online to an Orbitrap Exploris 480 (Thermo Fischer Scientific). Peptides were trapped for 5 min in 0.1% formic acid (FA) in water, using a 100-µm inner diameter 2-cm trap column (packed in-house with ReproSil-Pur C18-AQ, 3 µm) prior to separation on an analytical column (50 cm of length, 75 µM inner diameter; packed in-house with Poroshell 120 EC-C18, 2.7 µm). Trapped peptides were eluted following a 60 min gradient from 9–40% of 80% ACN, 0.1% FA. Full scan MS spectra from 350–1600 m/z were acquired in the Orbitrap at a resolution of 60,000 with a normalised AGC target of 300% and maximum injection time of 120 ms. Only peptides with charged states 3–8 were fragmented, and dynamic exclusion properties were set to $n = 1$, for a duration of 30 s. Fragmentation was performed using a stepped HCD collision energy mode (27, 30, 33%) in the ion trap and acquired in the Orbitrap at a resolution of 30,000 with an AGC target set to 500%, an isolation window of 1.4 m/z and a maximum injection time of 54 ms. Raw files obtained for respective IGX-MS experiments were subsequently analysed with pLink2[80]. FDR (controlled at PSM level for cross-linked spectrum matches and separately computed for intra and inter cross-links) rate was set to 5%. Acetylation (protein N-terminus), oxidation (Met) and phosphorylation (Ser/Thr/Tyr) were set as dynamic modifications. Carbamidomethylation (Cys) was set as a static modification.

Crosslinks observed in two out of three technical replicates for each sample were used for further analysis. A total of 41 cross-links were detected in unphosphorylated Mad1:C-Mad2 tetramer, and 68 and 62 cross-links for unphosphorylated and phosphorylated Mad1:C-Mad2:O-Mad2 hexamer, respectively. Crosslinks were plotted onto the Alphafold2 structures of tetrameric and hexameric Mad1:Mad2 complexes using the XMAS bundle in ChimeraX[81]. Venn diagrams showing the overlap between replicates and Boxen plots to visualise cross-link distances were generated using seaborn and matplotlib in python. The Boxen plots were created using the letter-value plot (Boxen plot) method[82]. Boxen plots are similar to a box plot, but they additionally plot the quartile values, starting with the median (Q2, 50th percentile) as the centreline, and each successive level outwards contains half of the remaining data (thereby describing the box bounds), continuing until the outlier level (5–8 outliers per tail) is reached. The box for each level out is depicted in lighter colour shades. A boxen plot was used to improve visualisation of the distribution of cross-link distances for Mad1:Mad2 complexes, which are highly flexible and thus have more outliers. The contact maps for highlighting the cross-links with respect to elongated and folded Mad1:C-Mad2:O-Mad2 were generated using the Bio.PDB package[83] in combination with seaborn and matplotlib in python.

## SEC-SAXS

The Mad1:Mad2 complexes (tetrameric Mad1$^{485–718}$:C-Mad2$^{R133A}$ and hexameric Mad1$^{485–718}$:C-Mad2:O-Mad2 in the unphosphorylated and phosphorylated states, where O-Mad2 is the Mad2$^{LL}$ [ref. 26] were analysed by size-exclusion chromatography coupled to small angle X-ray scattering (SEC-SAXS) at the BL21 beamline at Diamond Lightsource (DLS) (Harwell Campus, Didcot, United Kingdom), using a Shodex KW-403 column (Shodex) pre-equilibrated in 25 mM HEPES pH 7.5, 100 mM

NaCl, 1 mM TCEP[84]. To confirm column performance and instrument calibration, 40 µl of BSA at 10 mg/mL was analysed prior to experiments. In total, 40 µl of each sample was injected at 9, 6 or 3 mg/mL. No significant difference was seen between samples at different dilutions, and therefore only the highest concentration (9 mg/ml) of each sample is presented. Raw SAXS images were processed with the DAWN[85] processing pipeline at the DLS beamline to produced normalised and integrated 1-D un-subtracted SAXS curves. SEC-SAXS buffer subtraction and analyses were performed in Scatter IV (BioISIS). The expected Rg values of the modelled folded and elongated states of Mad1:Mad2 were calculated using the FoXS server[41].

## Computational methods

Molecular analyses were performed with the UCSF Chimera package[78]. Chimera is developed by the Resource for Biocomputing, Visualisation, and Informatics at the University of California, San Francisco (supported by NIGMS P41-GM103311). Molecular graphics were produced in PyMOL Molecular Graphics System, Version 2.3.3 Schrödinger, LLC. Protein structure predictions were generated using AlphaFold2[38] using the notebook-based ColabFold environment and the MMseqs2 MSA option[86]. The majority of protein secondary structure predictions were performed with PHYRE2 or PSIPRED proteins sequence analysis workbench[87–89]. Multiple sequence alignments were completed in BLAST[90].

## Reporting summary

Further information on research design is available in the Nature Research Reporting Summary linked to this article.

## Data availability

The data that support this study are available from the corresponding authors upon request. The NMR assignments of Cdc20[N] were deposited to the BMRB database (http://www.bmrb.wisc.edu/) with the accession number 51304. The in-gel cross-linking mass spectrometry (IGX-MS) data have been deposited to the ProteomeXchange Consortium via the PRIDE (https://www.ebi.ac.uk/pride/) partner repository with the dataset identifier PXD031872. PDBs used in this study: 4DZO, 1GO4, 2V64, 7B1F. Source data are provided with this paper.

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

## Acknowledgements

E.S.F. was funded by a Gates Cambridge Scholarship. This work was funded by an MRC grant (MC_UP_1201/6) and a CRUK grant (C576/A14109) to D.B. The XL-MS experiments were supported by the Horizon 2020 INFRAIA project Epic-XS (Project 823839 to A.J.R.H.). We thank K. Yan and the MRC-LMB EM facility for helping with EM data collection. J. Grimmett and T. Darling for computing and J. Shi for help with insect cell expression. We acknowledge S. Tamara (University of Utrecht) for additional cross-linking mass spectrometry experiments and T. Dendooven for advice in SEC-SAXS analyses. We thank Ajit Jogelkar and Andrea Musacchio for discussing unpublished work and for useful discussions regarding Mad1 fold-over and Mad2 conversion, and Andrea Musacchio for helpful comments on this manuscript. Some of the NMR studies were supported by the Francis Crick Institute through access to the MRC Biomedical NMR Centre. The Francis Crick Institute receives its core funding from Cancer Research UK (FC001029), the UK Medical Research Council (FC001029), and the Wellcome Trust (FC001029). The authors also thank the B21 SAXS beamline at the Diamond Light Source, Oxfordshire, for the measurements of mail-in SAXS samples.

## Author contributions

E.S.F. cloned, purified proteins, performed the biochemical assays, analysed SAXS data, collected and processed the cryo-EM data. C.W.H.Y. and S.M.V.F. acquired and analysed the NMR data. J.F.H. performed IGX-MS with supervision from A.J.R.H. S.H.M. completed AUC-SE and SEC-MALS. S.L.M. performed in-house mass spectrometry. E.S.F., C.W.H.Y., and D.B. wrote the manuscript with input from all authors.

## Competing interests

The authors declare no competing interests.
