## [Peer Review File · Nature Communications]

Juxtaposition of Bub1 and Cdc20 on phosphorylated Mad1 during catalytic mitotic checkpoint complex assemblyREVIEWER COMMENTS

Reviewer #1 (Remarks to the Author):

The manuscript from Fischer et al. employs an impressive array of structural and biophysical approaches to investigate how Mps1 phosphorylation of human Mad1 contributes to catalysis of mitotic checkpoint complex formation. The work builds on extensive prior in vitro and in vivo studies that have broadly outlined the mechanism that is delved into here. Overall, the study provides important mechanistic advances related to spindle assembly checkpoint signaling in human cells and is therefore well-suited for publication with revision.

General comments:

1) The manuscript has 3 elements, the first of which is focused on understanding precisely how the Cdc20 N-terminus interacts with phosphorylated C-terminus of Mad1 (with the primary phosphorylation site being Thr716). While the gold-standard here would be a co-crystal structure, the authors elegantly combine biochemical assays, NMR, and AlphaFold2 modeling to provide a picture of how this interaction occurs. Importantly, they explain how the interaction is compatible with the equally important association of phosphorylated Bub1 with the same C-terminal region of Mad1 (for which they had reported a co-crystal structure). This part of the manuscript is compelling and addresses an important question in the field (with the caveat that we are not qualified to evaluate NMR analysis). One potential formatting change that could strengthen this first element of the manuscript is to move the mutational analysis (along with the sequence alignments of the Cdc20 N-terminus) to a primary figure. These efforts represent the critical tests of the AlphaFold2 structural model and are far more important than the (admittedly, to non-experts) detailed summaries of NMR shifts. Featuring these validations more prominently would also help make this part of the manuscript, which represents a clear advance, very strong.

One minor point that should be addressed in the text, based on the model of the N-Cdc20 interaction with phosphorylated Mad1-CT, is whether the model explains why in the absence of Bub1 there is only 1 Cdc20 molecule bound to the dimeric Mad1. It would be good if this were described for a non-structural audience, as it is a striking property that is central to the tripartite mechanism.

A second related comment is whether pBub1 binding has any effect on the pMad1-N-Cdc20 interaction. With the current anisotropy assay employing a pBub1 peptide, affinity measurement of Mad1 CTD and Cdc20 N-terminus alone cannot not be done, so it is not essential to address this experimentally but perhaps could be commented on based on the model.

2) The second element of the manuscript relates to the folded state of the Mad1-Mad2 complex. This is potentially the most significant new insight reported here but also the most tenuous element of the manuscript. While we believe their claim that the complex is flexible, the cryo-EM in Fig. 5B does not lend any real confidence to a particular folded conformation, and the cross-linking mass spec traps and detects proximity (as acknowledged in the text). The states they show are likely among a range of possibilities and thus focusing on “a folded state” seems a little too strong – it would be more prudent to state that they have identified flexibility in the complex that may contribute to a potential mode of action. It is our understanding that when AlphaFold2 predicts a range of conformations differing at a hinge, it is an indication that there is flexibility in the region in question.

More importantly, there is no functional test of the significance of the complex’s flexibility, rather speculative rationalization of why it may be significant. While doing direct experimental analysis (e.g. replacing the predicted flexible region with a rigid coiled coil and addressing effect on catalysis) is not necessary for publication, it does require that the authors be circumspect in their description and statements on the significance of this flexibility.

3) The final element is the acceleration of Mad2 conformer conversion by MIM, which has been shown previously for the Mad1 MIM. This data reinforces that close access to the exposed MIM, potentially in an optimal geometric configuration achieved by the positioning mechanism, is key to the catalysis. This is not entirely surprising but it indicates that there is nothing inherently unusual about the Cdc20 MIM.

Minor comments:

1) Pg 8 “mutation of the conserved QYRL motif (Q648A/R650A) impaired binding to Mad1” should say “mutation of the conserved QYRL motif (Q648A/R650A) impaired binding to Cdc20”

2) Pg 9 “Because Mad1 phosphorylation is required for catalysing MCC formation but not Cdc20 kinetochore recruitment refs.14,33”, the correct references are 14 and 35.

Reviewer #2 (Remarks to the Author):

In this paper, Fischer and co-workers report the mechanistic studies of MCC formation through the assembly of a tripartite Bub1:Cdc20:Mad1 complex, which is regulated by sequential Mps1-dependent phosphorylation of Bub1 and Mad1-CTD. Using AlphaFold2, cryo-EM and cross-linking MS, the authors further proposed a Mad1-CTD 'fold-over' model to explain how the tripartite Bub1:Cdc20:Mad1 complex would position Cdc20-MIM in close proximity to O-Mad2 in the Mad1:C-Mad2:O-Mad2 hexamer. Extensive NMR studies have convincingly validated several key claims and conclusions. The proposed model is consistent with previously published biochemical and cellular studies. Overall, the manuscript includes some very nice work with lots of effort. The study provides significant insights into our understanding of C-Mad2:Cdc20 formation, a rate-limiting step in MCC assembly, by the Mps1-dependent pre-MCC catalytic scaffolding. However, there are some minor issues that need to be addressed before publication.

Specific points:

- 1) Page 11: 'Strikingly, in the presence of only a two-molar excess of a Cdc20-MIM peptide...in less than 30 mins'. There is nothing 'striking' here since O-Mad2 binding to MIM is instantaneously, much faster than the NMR detection window.
- 2) Figure 2: 1:4 molar ratio was used for 3a, but 1:2 molar ratio was used for 3b. Please clarify. Also, why A718 was used for normalization of peak intensity? It should use an N-terminal residue of Mad1 that is not involved in Cdc20 binding.
- 3) Figure 3: 1:4 molar ratio was used for 3a, but 1:2 molar ratio was used for 3b. Please clarify.
- 4) Figure 5b: the model fitting is not very convincing. There is a lot of 'empty' density for Mad1-CTD, and no explanation why no density at all for N-terminal helices of Mad1.
- 5) Figure 6 a and b: these experiments provide no new information on the kinetics of O-Mad2 to C-Mad2 conversion. They should be in supplemental figures.

Reviewer #3 (Remarks to the Author):

This paper presents an extensive structural study of hitherto uncharacterized elements of the Mitotic Checkpoint Complex assembly. It addresses the assemblies of the Mad1:Cdc20 and Bub1:Mad1 upon phosphorylation of Mad1 and Bub1 by Msp1. These interactions form the basis for the subsequent tripartite assembly of Cdc20, Bub1 and the C-terminal part of Mad1. The authors also find a state of the Mad1:Mad2 complex that gives hints at the mechanism of how the Mad1:Cdc20 complex then interacts

with Mad2 driven by Cdc20 catalyzing Mad2 to the closed form. Taken all these steps together, a model emerges as to how a pBub1:Cdc20:pMad1:C-Mad2:O-Mad2 complex is formed, and ultimately the MCC complex composed of Mad2, Cdc20, BubR1:Bub3.

This is a very well crafted and nicely presented study. The work is very detailed, well-rounded, convincing and made use of a broad range of structural and biophysical techniques (Mass Spec, SEC MALS, NMR, fluorescence, cryoEM). It is also an important contribution towards understanding the events leading to kinetochore interaction with microtubules and the spindle assembly checkpoint, where many aspects are not well understood yet.

I have a few minor questions that may be addressed:

- Has it been established that only the N-terminal part of Cdc20 interacts with MAD1?

- Is there a Mad1-C resonance assignment for the coiled-coil region (residues 597-610 or so)? I am a bit surprised that the coiled-coil resonance are visible.

- The effects of phosphorylation of Thr716 on Mad1 seem to be a bit contradictory: On the one hand, it has a strong function effect. On the other hand, some interactions quantified in this work are barely affected by it. For example, it seems that the main interaction region is 616-660. So why would then the phosphorylation cause such a difference in SEC-MALS? The authors may make a unifying statement considering all impact of the phosphorylation.

- Related to above: The AlphaFold tripartite complex is calculated without phosphorylation, yet the outcome seems representative of the phosphorylated case (so no effect?); can the authors mimic the phosphorylation with an aspartic acid substitution?

- Figure S3: The Cdc20 Box1 causes minimal chemical shift changes on Mad1; does it change intensities (that is hard to see because of the near-perfect superposition of most of the peaks)?

- Figure 3e: Why are the Cb chemical shift not used to help predict the secondary structure?

Reviewers' comments:

Referees' comments are coloured blue, our responses are coloured black, and major changes to the text are copied here and coloured red.

Reviewer #1 (Remarks to the Author):

The manuscript from Fischer et al. employs an impressive array of structural and biophysical approaches to investigate how Mps1 phosphorylation of human Mad1 contributes to catalysis of mitotic checkpoint complex formation. The work builds on extensive prior in vitro and in vivo studies that have broadly outlined the mechanism that is delved into here. Overall, the study provides important mechanistic advances related to spindle assembly checkpoint signaling in human cells and is therefore well-suited for publication with revision.

General comments:

1) The manuscript has 3 elements, the first of which is focused on understanding precisely how the Cdc20 N-terminus interacts with phosphorylated C-terminus of Mad1 (with the primary phosphorylation site being Thr716). While the gold-standard here would be a co-crystal structure, the authors elegantly combine biochemical assays, NMR, and AlphaFold2 modeling to provide a picture of how this interaction occurs. Importantly, they explain how the interaction is compatible with the equally important association of phosphorylated Bub1 with the same C-terminal region of Mad1 (for which they had reported a co-crystal structure). This part of the manuscript is compelling and addresses an important question in the field (with the caveat that we are not qualified to evaluate NMR analysis).

One potential formatting change that could strengthen this first element of the manuscript is to move the mutational analysis (along with the sequence alignments of the Cdc20 N-terminus) to a primary figure. These efforts represent the critical tests of the AlphaFold2 structural model and are far more important than the (admittedly, to non-experts) detailed summaries of NMR shifts. Featuring these validations more prominently would also help make this part of the manuscript, which represents a clear advance, very strong.

We thank the reviewer for their positive comments and helpful suggestions to improve our manuscript.

Response: We thank the reviewer for this excellent idea to strengthen our manuscript. Based on the reviewer's suggestions we have moved the mutational analyses of our AlphaFold2 model from the supplementary section to a main figure (now Figure 5) and we also included a sequence alignment of the Cdc20 N-terminal residues 1-40 in this figure, with asterisks marking the specific residues within α 1 and Box1 tested by SEC-MALS and FP.

One minor point that should be addressed in the text, based on the model of the N-Cdc20 interaction with phosphorylated Mad1-CT, is whether the model explains why in the absence of Bub1 there is only 1 Cdc20 molecule bound to the dimeric Mad1. It would be good if this were

described for a non-structural audience, as it is a striking property that is central to the tripartite mechanism.

Response: We thank the reviewer for bringing up this important point. To address this point, in our previous discussion of Cdc20 stoichiometry on pg 6, we have further emphasized that the structure of Mad1^{CTD} is asymmetric as observed in apo-Mad1^{CTD} and Bub1^{CD1}-Mad1^{CTD} crystal structures. “We previously showed that only a single Bub1^{CD1} binds to the Mad1^{CTD} homodimer, most likely a result of the inherent asymmetry within Mad1^{CTD} in which the coiled-coil is bent with respect to the head domain, **observed in both the Bub1^{CD1} bound and apo-Mad1^{CTD} X-ray structures in multiple different crystal lattices^{21,24}**. Interestingly, the Mad1^{CTD} homodimer also binds only one Cdc20^N (Fig 2b and Supp Fig 2). We reasoned that Mad1^{CTD} asymmetry might also play a role in defining the Cdc20^N:Mad1^{CTD} stoichiometry. As shown by NMR, residues of Mad1^{CTD} that bind Cdc20^N almost entirely overlap in sequence with those that bind Bub1^{CD1} (Fig 2c-g)²¹.”

A second related comment is whether pBub1 binding has any effect on the pMad1-N-Cdc20 interaction. With the current anisotropy assay employing a pBub1 peptide, affinity measurement of Mad1 CTD and Cdc20 N-terminus alone cannot not be done, so it is not essential to address this experimentally but perhaps could be commented on based on the model.

Response: We agree with the reviewer that testing whether pBub1 binding has an effect on the pMad1^{CTD}:Cdc20^N interaction and vice versa is an important and interesting question. As the reviewer mentions, although our current anisotropy assay does not allow us to directly measure Cdc20^N binding to pMad1^{CTD} alone, we have now tested binding of AF488-Bub1^{CD1} to preformed Cdc20^N:pMad1^{CTD} and found that this gives a very similar K_D to that of AF488-Bub1^{CD1} binding to pMad1^{CTD} alone. This experiment shows that the binding of Bub1^{CD1} to Mad1^{CTD} is the same regardless of whether or not Mad1^{CTD} is in complex with Cdc20. This indicates there is no cooperativity between Bub1 and Cdc20 binding to pMad1^{CTD}. It then follows that the binding affinity of Cdc20^N to Mad1^{CTD} is unaffected by whether Mad1^{CTD} is in complex with Bub1^{CD1}.

We have now included this experiment in Fig 4c, in the text of our manuscript on pg 7 and methods section on pg 18.

Fig 4c legend: “**Fluorescence anisotropy measurements of AF488-Bub1^{CD1} binding to preformed pMad1^{CTD}:Cdc20^N, where the concentration of Cdc20^N is kept constant (23 μM) and Mad1^{CTD} is titrated, giving a K_D of 1.7 μM. This affinity is similar to that of AF488-Bub1^{CD1} binding to pMad1^{CTD} alone (0.9 μM) and suggests the binding of Bub1^{CD1} and Cdc20^N to pMad1^{CTD} is unlikely to be cooperative.**”

“**We next determined the affinity of AF488-Bub1^{CD1} for a preformed pMad1^{CTD}:Cdc20^N complex. This showed a similar affinity (1.7 μM) to the binding of AF488-Bub1^{CD1} to pMad1^{CTD} alone (0.9 μM), indicating that Cdc20 and Bub1 binding to pMad1^{CTD} is not cooperative (Fig 4c).**”

“To analyse binding of Bub1^{CD1} to the Mad1^{CTD}:Cdc20^N complex and test for cooperativity, a 2-fold dilution series of Mad1^{CTD} from 96 μM to 2.9 nM was mixed 1:1 with 92 μM of Cdc20^N, incubated on ice for 30 mins and then mixed 1:1 with 40 nM of AF488-Bub1^{CD1}.”

2) The second element of the manuscript relates to the folded state of the Mad1-Mad2 complex. This is potentially the most significant new insight reported here but also the most tenuous element of the manuscript. While we believe their claim that the complex is flexible, the cryo-EM in Fig. 5B does not lend any real confidence to a particular folded conformation, and the cross-linking mass spec traps and detects proximity (as acknowledged in the text). The states they show are likely among a range of possibilities and thus focusing on “a folded state” seems a little too strong – it would be more prudent to state that they have identified flexibility in the complex that may contribute to a potential mode of action. It is our understanding that when AlphaFold2 predicts a range of conformations differing at a hinge, it is an indication that there is flexibility in the region in question.

Response: We thank the reviewer for these insightful suggestions regarding our use of ‘a folded state’. We agree with the reviewer that we have identified how the flexible linker allows Mad1^{CTD} to fold-over therefore bringing the head domain close to the Mad1:C-Mad2:O-Mad2 core, but this Mad1^{CTD} fold-over remains highly dynamic and does not seem to adopt a specific or stable folded state. We have now altered the presentation and discussion of our cryo-EM and cross-link mass spectrometry data to ensure that we are not concluding that there is ‘a specific folded state’. See pages 9-11 for the various alterations.

More importantly, there is no functional test of the significance of the complex’s flexibility, rather speculative rationalization of why it may be significant. While doing direct experimental analysis (e.g. replacing the predicted flexible region with a rigid coiled coil and addressing effect on catalysis) is not necessary for publication, it does require that the authors be circumspect in their description and statements on the significance of this flexibility.

Response: We agree with the reviewer that we did not perform a functional test of the significance of the complex’s flexibility and of the ability of Mad1^{CTD} to fold-over. While our manuscript was under revision, Ajit Jogelkar, Andrea Musacchio and colleagues, submitted a study on BioRxiv (Chen et al., 2022), which identifies existence of the folded state *in vivo* and evaluates the functional significance of this flexible linker (which they refer to as a hinge) within the Mad1 C-terminus and the ability of Mad1^{CTD} to fold-over. They also replaced the flexible region with a rigid coiled-coil and identify impairment on catalytic MCC formation. We have now cited and discussed this study in our revised manuscript.

On pg 10: “More recently, fold-over of Mad1 has also been detected *in vivo* using fluorescence-lifetime imaging (FLIM)³⁹.”

On pg 13: “In a more recent complementary study, Chen and colleagues demonstrate that the flexible hinge within Mad1 which allows Mad1^{CTD} fold-over is essential for catalytic MCC

assembly *in vitro* and SAC signalling *in vivo*³⁹. They also identified that catalysis induced by the flexible linker was dependent upon an intact Mad1:Mad2 interaction with Bub1^{CD1} and Cdc20^{Box1}, thereby coupling Mad1 flexibility and tripartite Bub1^{CD1}:Cdc20^N:Mad1^{CTD} assembly with catalytic Mad2:Cdc20 formation.”

3) The final element is the acceleration of Mad2 conformer conversion by MIM, which has been shown previously for the Mad1 MIM. This data reinforces that close access to the exposed MIM, potentially in an optimal geometric configuration achieved by the positioning mechanism, is key to the catalysis. This is not entirely surprising but it indicates that there is nothing inherently unusual about the Cdc20 MIM.

Minor comments:

1) Pg 8 “mutation of the conserved QYRL motif (Q648A/R650A) impaired binding to Mad1” should say “mutation of the conserved QYRL motif (Q648A/R650A) impaired binding to Cdc20”

Response: We thank the reviewer for catching this typo. We have now corrected this in the text.

2) Pg 9 “Because Mad1 phosphorylation is required for catalysing MCC formation but not Cdc20 kinetochore recruitment refs.14,33”, the correct references are 14 and 35.”

Response: We thank the reviewer for spotting this error. We have now fixed this in the text.

Reviewer #2 (Remarks to the Author):

In this paper, Fischer and co-workers report the mechanistic studies of MCC formation through the assembly of a tripartite Bub1:Cdc20:Mad1 complex, which is regulated by sequential Mps1-dependent phosphorylation of Bub1 and Mad1-CTD. Using AlphaFold2, cryo-EM and cross-linking MS, the authors further proposed a Mad1-CTD ‘fold-over’ model to explain how the tripartite Bub1:Cdc20:Mad1 complex would position Cdc20-MIM in close proximity to O-Mad2 in the Mad1:C-Mad2:O-Mad2 hexamer. Extensive NMR studies have convincingly validated several key claims and conclusions. The proposed model is consistent with previously published biochemical and cellular studies. Overall, the manuscript includes some very nice work with lots of effort. The study provides significant insights into our understanding of C-Mad2:Cdc20 formation, a rate-limiting step in MCC assembly, by the Mps1-dependent pre-MCC catalytic scaffolding. However, there are some minor issues that need to be addressed before publication.

We thank the reviewer for positive comments and helpful suggestions to improve the manuscript.

Specific points:

1) Page 11: 'Strikingly, in the presence of only a two-molar excess of a Cdc20-MIM peptide...in less than 30 mins'. There is nothing 'striking' here since O-Mad2 binding to MIM is instantaneously, much faster than the NMR detection window.

Response: In Figure 7a and b (previously Figure 6a and b), we systematically compared the conversion rate of O-Mad2 to C-Mad2 in the presence or absence of Cdc20^{MIM} and our data show a significant difference in the conversion rates (30 minutes versus 24 hours). This is important to illustrate that the key kinetic energy barrier to MCC formation is readily overcome by promoting the accessibility and repositioning of Cdc20^{MIM} for Mad2 binding. In these experiments we are using NMR to monitor the conversion of O-Mad2 to C-Mad2 bound to Cdc20^{MIM}. Thus, the changes in the NMR spectra are reporting the conformational change of O-Mad2 to Cdc20^{MIM} bound C-Mad2. These changes are not monitoring a simple bi-molecular interaction of Cdc20^{MIM} to either O- or C-Mad2. In our experimental conditions, the conversion of Mad2 takes tens of minutes in the presence of Cdc20^{MIM}, and is within the detection window of NMR. This compares to the extremely slow rate of conversion of O-Mad2 to C-Mad2 in the absence of Cdc20^{MIM} (hours). It is unknown whether Cdc20^{MIM} binds to O-Mad2 (which would be a spontaneous event). We could observe a mixture of O-Mad2 and C-Mad2 in the first 15 minutes upon addition of Cdc20^{MIM}, which indicates the conversion took more than 15 minutes to complete under these conditions. We have now added an additional panel to Figure 7b (previously Figure 6b) as well as to Supp Figure 13 (previously Figure 14) to highlight the 15-minute time point and improve clarity to reviewers.

We also thank the reviewer for pointing out that our use of the word 'striking' might be too subjective, we have therefore removed it in the text.

2) Figure 2: 1:4 molar ratio was used for 3a, but 1:2 molar ratio was used for 3b. Please clarify.

Response: Because a 1:2 molar ratio was used in both Figure 3a and 3b, we think the reviewer is asking about the difference of molar ratio used in Figure 2 versus Figure 3, where a maximum of 1:4 Cdc20^N was titrated into isotopically labelled Mad1^{CTD} (Figure 2) and a maximum of 1:2 Mad1^{CTD} was titrated into isotopically labelled Cdc20^N (Figure 3).

To identify the binding sites of Mad1^{CTD} on Cdc20^N, we used ¹³C-detected 2D CON spectra that report the correlation between backbone carbonyl ¹³C and amide ¹⁵N, to include resonances from prolines and any other solvent exchanged residues that are absent in ¹H,¹⁵N 2D HSQC (Supp Figure 5). However, ¹³C-detected experiments have the disadvantage of a lower sensitivity and therefore we were limited from using a more dilute sample with further excess of Mad1^{CTD}. 200 μM Cdc20^N in the presence of 400 μM Mad1^{CTD} was used to collect these spectra, given a K_D in the range of 5 – 30 μM (Figure 4b), over 90% of Cdc20^N in the sample is Mad1^{CTD}-bound and should therefore provide a reliable analysis of the binding sites.

Also, why A718 was used for normalization of peak intensity? It should use an N-terminal residue of Mad1 that is not involved in Cdc20 binding.

Response: We thank the reviewer for pointing out this lack of clarity. We have included further clarification in the figure legend of Figure 2e. “Peak intensities were normalized to the C-terminal residue Ala718 for the spectra of unphosphorylated Mad1^{CTD}, where the C-terminus was clearly not involved in binding. For phosphorylated Mad1^{CTD} the peak intensities were adjusted to match the spectra collected for unphosphorylated Mad1^{CTD} for comparison, as the experimental conditions were nearly identical.”

4) Figure 5b: the model fitting is not very convincing. There is a lot of ‘empty’ density for Mad1-CTD, and no explanation why no density at all for N-terminal helices of Mad1.

Response: We thank the reviewer for these comments. We agree that the fitted model shown in Figure 5b does not account for all the EM density. As we point out in the text this is only a ~16 Å reconstruction at FSC = 0.5. To lessen the focus our manuscript places on this fitted model we have now placed it in Supp Figure 11 (now Supp Figure 10). We also agree that there is ‘empty’ density for the CTD of Mad1. We suspect this is because the head domain is highly flexible, and the particles used for the reconstruction still encompass the head domain captured in a variety of folded positions and therefore the reconstructed density of this region becomes an average of this flexibility. There are several possible explanations for why the N-terminal helices disappear in our reconstruction of the folded conformation of Mad1 and are also missing in the 2D averages shown in Figure 6a (previously Fig 5a). We think most probable is because of how the folded-state particles orient on the cryo-EM grid, which may obscure the N-terminal helices. It may also be that the folded state somehow perturbs the N-terminal coiled-coil, making it more flexible and thus less visible by cryo-EM.

We have now updated our manuscript to include a brief discussion of these points on page 10.

“More recently, fold-over of Mad1 has also been detected *in vivo* using fluorescence-lifetime imaging (FLIM)³⁹. The small size and conformational variability of this complex, despite been cross-linked, precluded a high-resolution cryo-EM reconstruction and suggests there is no stable folded state of Mad1^{CTD} but that Mad1^{CTD} remains highly dynamic (Supp Fig 10a,b). A medium-resolution 3D reconstruction (11 Å at FSC = 0.143 or 16 Å at FSC = 0.5) of a folded state of Mad1 allowed the fitting of crystal structures^{21,23,24,26} and visualization of Mad1^{CTD} positioned next to the core (Supp Fig 10d,e,g). The presence of ‘empty’ density around Mad1^{CTD} in this reconstruction is likely from this subset of particles containing Mad1^{CTD} in various orientations. Density for the N-terminal helices was not recovered, which may be due to how the folded particles adhere to the EM-grid, or due to Mad1^{CTD} fold-over increasing their flexibility.”

5) Figure 6 a and b: these experiments provide no new information on the kinetics of O-Mad2 to C-Mad2 conversion. They should be in supplemental figures.

Response: We thank the reviewer for this suggestion and apologize that our presentation of the experiments in Figure 7a and b (previously Figure 6a and b) lacked clarity. Upon reflection we decided that calling Cdc20 MIM a catalyst for the open-to-closed conversion of Mad2 might have caused confusion. While a catalyst should not alter the product of a reaction, the empty C-

Mad2 and Cdc20^{MIM}-bound C-Mad2 adopt distinctly different conformations (Figure 7a). This implies that Cdc20^{MIM} is not a catalyst that simply promotes the rate of a reaction, instead it binds and stabilizes C-Mad2 to give a different product. To support this statement, we have conducted experiments using a sub-stoichiometric concentration of Cdc20^{MIM} to O-Mad2 (1:50) and found that nearly all of Mad2 remains in an open state (Supp Fig 13c), in stark contrast to the spontaneous conversion observed when Cdc20^{MIM} was added in a 2-fold excess (Supp Fig 13a).

However, we still feel a summary of these experiments is worthy of inclusion in Figure 7 rather than in the supplementary section because they demonstrate how the Cdc20^{MIM} peptide induces conversion and stabilizes bound C-Mad2, similar to Mad1^{MIM}. We have now restructured Figure 7a and b and Supp Figure 13, to emphasize the importance of the catalytic platform for MCC assembly. We have also clarified our explanation of this in the manuscript and avoided calling Cdc20^{MIM} a catalyst. Our results emphasize how the presentation of Cdc20^{MIM} alone is sufficient to trigger Mad2 conversion and highlights that the main role of the catalytic platform is to spatially reposition Cdc20^{MIM} for its interaction with O-Mad2, therefore contribute to overcoming a key kinetic energy barrier to MCC assembly.

Updated figure legend: **“Supp Fig 13: The MIM of Cdc20 induces Mad2 conversion.** ¹H, ¹⁵N 2D HSQC of ¹⁵N-labelled O-Mad2 during open-to-closed conversion of Mad2. **a,b** Conversion of 100 μM ¹⁵N-labelled O-Mad2 R133A in the presence (a) or absence (b) of 200 μM Cdc20^{MIM} peptide at 25°C. As shown in Figure 7a, the side chains of Trp167 and Trp75 are diagnostic of Mad2 conformation, and their resonances are boxed in the spectra. The schematics show the conformation of Mad2 as indicated by the tryptophan side chain resonances. **c** Substoichiometric concentration of Cdc20^{MIM} was used in (c) where 2 μM of Cdc20^{MIM} was added to 100 μM ¹⁵N-labelled O-Mad2 R133A. The majority of Mad2 remains in an open state, in marked contrast to the spontaneous conversion observed when Cdc20^{MIM} was added in a 2-fold excess (a). This suggests that Cdc20^{MIM} cannot be reused in the reaction and is therefore not a catalyst for Mad2 conversion. Instead Cdc20^{MIM} functions to induce conversion and stabilizes Cdc20^{MIM}-bound C-Mad2.”

Updated text: **“We also showed that such spontaneous conversion cannot be achieved using a substoichiometric concentration of Cdc20^{MIM} (1:50 to Mad2, Supp Fig 13c). This suggests that Cdc20^{MIM} functions to trigger conversion and stabilizes Cdc20^{MIM}-bound C-Mad2. The capacity of Cdc20^{MIM} to induces the O-Mad2 to C-Mad2 conversion is reminiscent of the O-Mad2 to C-Mad2 conversion induced by Mad1^{MIM} [ref 29].”**

Reviewer #3 (Remarks to the Author):

This paper presents an extensive structural study of hitherto uncharacterized elements of the Mitotic Checkpoint Complex assembly. It addresses the assemblies of the Mad1:Cdc20 and Bub1:Mad1 upon phosphorylation of Mad1 and Bub1 by Msp1. These interactions form the basis for the subsequent tripartite assembly of Cdc20, Bub1 and the C-terminal part of Mad1.

The authors also find a state of the Mad1:Mad2 complex that gives hints at the mechanism of how the Mad1:Cdc20 complex then interacts with Mad2 driven by Cdc20 catalyzing Mad2 to the closed form. Taken all these steps together, a model emerges as to how a pBub1:Cdc20:pMad1:C-Mad2:O-Mad2 complex is formed, and ultimately the MCC complex composed of Mad2, Cdc20, BubR1:Bub3.

This is a very well crafted and nicely presented study. The work is very detailed, well-rounded, convincing and made use of a broad range of structural and biophysical techniques (Mass Spec, SEC MALS, NMR, fluorescence, cryoEM). It is also an important contribution towards understanding the events leading to kinetochore interaction with microtubules and the spindle assembly checkpoint, where many aspects are not well understood yet.

I have a few minor questions that may be addressed:

1) Has it been established that only the N-terminal part of Cdc20 interacts with MAD1?

We thank the reviewer for positive comments and helpful suggestions to improve our manuscript.

Response: The available evidence in the literature suggests that only the N-terminus of Cdc20 interacts with Mad1. Hongtao Yu's lab (Ji et al, 2017), reported that truncations of Cdc20 lacking the first 60 amino acids (Cdc20^{ΔN60}) did not interact with phosphorylated Mad1⁴⁸⁵:Mad2 (Figure 6 of their paper), suggesting that only the N-terminus of Cdc20 is involved. Cross-link mass spectrometry studies by Andrea Musacchio's lab (Piano et al, 2021) also suggested only the N-terminus (specifically Box1) was involved (Figure 1A in their paper). Similar findings in *C. elegans* have also been reported (Lara-Gonzalez et al, 2021). In our study we use a truncation encompassing only the first 73 amino acids of Cdc20, which was used in the Piano et al, 2021 study, and we have cited this study correspondingly.

2) Is there a Mad1-C resonance assignment for the coiled-coil region (residues 597-610 or so)? I am a bit surprised that the coiled-coil resonance are visible.

Response: Yes, we have assigned the backbone resonances of Mad1^{CTD} (residues 597-718) in a previous study (Fischer et al., 2021) and this was referred to in our manuscript. A uniformly sidechain perdeuterated sample together with relaxation optimized 3D experiments were used for assignment to counteract the reduction in sensitivity associated with the slow overall tumbling of the elongated Mad1^{CTD} dimer.

- The effects of phosphorylation of Thr716 on Mad1 seem to be a bit contradictory: On the one hand, it has a strong function effect. On the other hand, some interactions quantified in this work are barely affected by it. For example, it seems that the main interaction region is 616-660. So why would then the phosphorylation cause such a difference in SEC-MALS? The authors may make a unifying statement considering all impact of the phosphorylation.

Response: We thank the reviewer for this comment, and we would like to elaborate on the importance of phosphorylation of Thr716 on Mad1:Cdc20 binding. Our NMR titration data show that $\alpha 1$ interacts with residue 616-660 of Mad1^{CTD} while Box1 specifically interacts with phosphorylated pT716 and the head domain (Supp Fig 6). Notably the interaction between pMad1^{CTD}:Cdc20^{Box1} appeared stronger than that between pMad1^{CTD}:Cdc20 ^{$\alpha 1$} , as a higher concentration of Cdc20 $\alpha 1$ peptide was required to observe significant chemical shift perturbations by NMR. These interactions were further illustrated in our AlphaFold2 model (Figure 4c) where multiple positively charged residues on Cdc20 Box1 can make extensive interactions with the negatively charged pT716. Our data suggests that pT716 phosphorylation is essential for the stronger binding of Cdc20 Box1 to pMad1^{CTD}, while Cdc20 $\alpha 1$ has a weaker binding to pMad1^{CTD} that is independent of phosphorylation. Cooperatively this leads to a tight binding (K_D : 4.8 μ M) between pMad1^{CTD} and Cdc20^N, and a weaker binding (K_D : 28.1 μ M) in the absence of phosphorylation (Figure 4b).

- Related to above: The AlphaFold tripartite complex is calculated without phosphorylation, yet the outcome seems representative of the phosphorylated case (so no effect?); can the authors mimic the phosphorylation with an aspartic acid substitution?

Response: We thank the reviewer for their insightful question. Recent studies have suggested that even though AlphaFold2 does not include modifications, ligands or cofactors in the prediction, it can generate models with space to accommodate these additions for sequences with extensive multiple sequence alignment (MSA) (Bagdonas et al., 2021; Hekkelman et al., 2021).

In our manuscript we mention on pg 8 (with the red text highlighted being altered to provide further clarity) “The AlphaFold2 prediction was generated without including a phosphorylated Thr716 in the input sequence. We assume that Thr716 of Mad1 co-evolves with the basic residues of Box1, and the multisequence alignment (MSA) algorithm of AlphaFold2 likely creates a close distance constraint between these residues³⁸. This is supported by the accurate prediction of the Bub1^{CD1}:Mad1^{CTD} interaction despite the sequence of Bub1^{CD1} not containing a phosphorylated Thr461 which is essential for the Bub1^{CD1}:Mad1^{CTD} interaction^{14,20,21}.”

As our AlphaFold2 model is generated using MSA, a single Thr to Asp mutation will not severely impact the results of the prediction. Nevertheless, we did test this hypothesis by running AlphaFold2 with this substitution and found no significant change in the prediction (data not shown).

Additionally, we do not see the need to test a phosphomimetic mutation *in vitro* as we are able to obtain a near complete phosphorylation of Mad1 Thr716 using purified full-length Mps1 kinase, as illustrated by our MS and NMR data (Supp Fig 1).

- Figure S3: The Cdc20 Box1 causes minimal chemical shift changes on Mad1; does it change intensities (that is hard to see because of the near-perfect superposition of most of the peaks)?

Response: We thank the reviewer for pointing this out. We did not observe any major changes in peak intensities (see embedded figure below). This agrees with the weak binding (likely in the mM range) we observed between Cdc20^{Box1} and Mad1^{CTD}, which is in a fast exchange regime of NMR timescale. To clarify this further in the text we added the statement “**Only chemical shift perturbations are shown because no significant change in peak intensity was observed.**” to the legend of Supp Fig 6.

Figure: Relative peak intensities of ¹⁵N-labelled Mad1^{CTD} after titration of Cdc20^{Box1} in a ¹H, ¹⁵N 2D HSQC.

- Figure 3e: Why are the C α chemical shifts not used to help predict the secondary structure?

Response: The use of secondary C α chemical shifts has been shown to provide accurate predictions of secondary structure propensities, in fact it has been shown that C α chemical shifts have higher reliability in distinguishing α -helices from random coils (Tamiola and Mulder, 2012). We have also noticed that to identify secondary structure propensities in a largely disordered protein like Cdc20^N, it is critical to have a good random coil chemical shift reference and our approach of using the C α chemical shifts from a urea-denatured state as reference has provided a sensitive detection of any residual secondary structure. This has also been illustrated by Ad Bax’s lab in their recent publication (Kakeshpour et al., 2021).

References:

Bagdonas, H., Fogarty, C. A., Fadda, E. & Agirre, J. The case for post-predictional modifications in the AlphaFold Protein Structure Database. *Nat. Struct. Mol. Biol.* **28**, 869–870 (2021).

Chen, C. et al. The Structural Flexibility of MAD1 Facilitates the Assembly of the Mitotic Checkpoint Complex. *bioRxiv* 2022.06.29.498198 (2022). doi:10.1101/2022.06.29.498198

Hekkelman, M. L., Vries, I. de, Joosten, R. P. & Perrakis, A. AlphaFill: enriching the AlphaFold models with ligands and co-factors. *bioRxiv* 2021.11.26.470110 (2021). doi:10.1101/2021.11.26.470110

Tamiola, K. & Mulder, F. A. A. Using NMR chemical shifts to calculate the propensity for structural order and disorder in proteins. *Biochem. Soc. Trans.* **40**, 1014–1020 (2012).

Kakeshpour, T. *et al.* A lowly populated, transient β -sheet structure in monomeric A β 1-42 identified by multinuclear NMR of chemical denaturation. *Biophys. Chem.* **270**, 106531 (2021).

REVIEWERS' COMMENTS

Reviewer #1 (Remarks to the Author):

The authors have done an excellent job of revising an already-strong manuscript. It should be accepted for publication.

Reviewer #2 (Remarks to the Author):

The authors have addressed all my comments. I support publication of this manuscript.

Reviewer #3 (Remarks to the Author):

All my raised concerns have been addressed in this revised version, as well as those of the other reviewers.

I would like to restate that this work has been executed very carefully and is well presented, and is also relevant to the field. therefore, I recommend it be published in Nature Communications in the present form.